# Utilizing a Multi-Proxy to Model Comparison to Constrain the Season and Regionally Heterogeneous Impacts of the Mt. Samalas 1257 Eruption

Laura Wainman[1,2], Lauren R Marshall[3,4], Anja Schmidt[3,5,6]

[1]Department of Earth Sciences, University of Cambridge, Cambridge, CB2 3EQ, UK
[2]School of Earth and Environment, University of Leeds, Leeds, LS2 9JT, UK
[3]Yusuf Hamied Department of Chemistry, Cambridge, CB2 1EW, UK
[4]Department of Earth Sciences, Durham University, Durham, DH1 3LE, UK
[5]Institute of Atmospheric Physics (IPA), German Aerospace Centre (DLR), Oberpfaffenhofen, 82234, Germany
[6]Meterological Institute, Ludwig Maximilian University of Munich, Munich, 80333, Germany

*Correspondence to*: Laura Wainman (eelrw@leeds.ac.uk)

**Abstract.** The Mt. Samalas eruption, thought to have occurred in Summer 1257, ranks as one of the most explosive sulfur-rich eruptions of the Common Era. Despite recent convergence, several dates have been proposed for the eruption ranging between 1256-1258. As of yet, no single combination of evidence has been able to robustly distinguish between, and exclude the other dates proposed for the Mt Samalas eruption. Widespread surface cooling and hydroclimate perturbations following the eruption have been invoked as contributing to a host of 13th Century social and economic crises, although regional-scale variability in the post-eruption climate response remains uncertain. In this study we run ensemble simulations using the UK Earth System Model (UKESM1) with a range of eruption scenarios and initial conditions in order to compare our simulations with a globally-resolved multi-proxy database for the Mt. Samalas eruption, incorporating tree-rings, ice cores, and historical records. This allows precise constraints to be placed on the year and season of the Mt. Samalas eruption as well as an investigation into the regionally heterogeneous post-eruption climate response. Using a multi-proxy to model comparison, we are able to robustly distinguish between July 1257 and January 1258 eruption scenarios where the July 1257 ensemble simulation achieves considerably better agreement with spatially averaged and regionally resolved proxy surface temperature reconstructions. These reconstructions suggest the onset of significant cooling across Asia and Europe in early 1258, and thus support the plausibility of previously inferred historical connections. Model-simulated temperature anomalies also point to severe surface cooling across the Southern Hemisphere with as of yet unexplored historical implications for impacted civilizations. Model simulations of polar sulfate deposition also reveal distinct differences in the timing of ice sheet deposition between the two simulated eruption dates, although comparison of the magnitude or asymmetric deposition of sulfate aerosol remains limited by large inter-model differences and complex intra-model dependencies. Overall, the multi-proxy to model comparison employed in this study has strong potential in constraining similar uncertainties in eruption source parameters for other historical eruptions where sufficient coincident proxy records are available, although care is needed to avoid the pitfalls of model-multi proxy comparison.

## 1 Introduction

The Mt. Samalas eruption, which occurred on the Indonesian Island of Lombok between 1257 and 1258, is identified in ice cores as one of the largest volcanic sulfate deposition events of the last 2500 years (Palais et al., 1992, Zielinski et al., 1994, Sigl et al., 2015). Petrological analysis suggests a release of ~120Tg of sulfur dioxide (SO2) into the stratosphere, with a maximum estimated plume height of 43 km and volcanic explosivity index (VEI) of 7, meaning the eruption ranks as one of the most-explosive sulfur-rich eruptions of the Common Era (Lavigne et al., 2013, Vidal et al., 2015, Toohey and Sigl, 2017). Tree-ring reconstructions suggest a peak Northern Hemisphere (NH) average summer cooling of -0.8°C to -1.2°C between Summer 1258 and 1259 (Schneider et al., 2015, Guillet et al., 2017, Wilson et al., 2016, Büntgen et al,. 2022). The surface air temperature (SAT) anomalies and potential hydroclimate perturbations induced by the eruption have been invoked by historians as contributing factors to a host of 13th Century social and economic crises (Campbell, 2017, Malawani et al., 2022, Guillet et al., 2017, Bierstedt, 2019, Stothers 2000, Green 2020, Di Cosmo et al., 2021).

The full span of dates proposed for the Mt Samalas eruption ranges from 1256 to 1258, with suggestions including an eruption in spring 1256 (Bauch, 2019), summer 1257 (Lavigne et al., 2013, Oppenheimer 2003), and early 1258 (Stothers 2000). Whilst consensus has converged on a summer 1257 eruption date, as of yet, no single combination of evidence has been able to robustly distinguish between, and exclude other dates proposed for the Mt Samalas eruption. The suggestion of Spring 1256 was based on historical evidence for a dust veil over Asia and the Middle East in late 1256 and early 1257 (Bauch, 2019). This is more likely attributed to a smaller eruption such as the 1256 Medina eruption which had only localized impacts (Saliba, 2017). A mid-1257 eruption date was first proposed by Oppenheimer (2003) based on the spatial distribution of negative temperature anomalies across both hemispheres for 1257-59. Radiocarbon dating of the pyroclastic flow deposits associated with the eruption also yield a youngest eruption age boundary of 1257, with some samples being older, but no samples being younger than 1257 (Lavigne et al., 2013). Based on the westerly displacement of ash isopachs, Lavigne et al., (2013) proposed that easterly winds prevailed at the time of the eruption, indicative of the eruption occurring during the May-Oct dry season. Negative tree-ring width (TRW) growth anomalies in the late 1257-growth season (Büntgen et al., 2022) and frost rings (Salzer and Hughes, 2007) in the Western US in 1257 and 1259 also add support for a potential eruption date prior to August 1257. Modelling studies for the Mt. Samalas eruption have achieved best agreement with tree-ring reconstructions for a May-July eruption window (Stoffel et al., 2015). Nonetheless, Stothers (2000) suggests a later eruption date of early 1258 based on peak sulfate deposition for the Mt. Samalas eruption occurring in 1259 (from Hammer et al., 1980) and the first historical reports of a dust veil over Europe appearing in Summer 1258, which they suggest is most compatible with an early 1258 eruption. Therefore, there is still a need to constrain the year and season of the eruption with greater certainty, with implications for evaluating the robustness of inferred connections to synchronous historical events, as well as in the role of the eruption sulfate deposition spike as a key temporal calibration marker in ice core records.

The regionally heterogeneous climate response to the Mt. Samalas eruption also remains largely unconstrained at a global level. Guillet et al., (2017) utilized a wealth of historical records and tree-ring chronologies to assess the impact of the eruption across the NH, with particular focus on the climate response to the eruption revealed by historical sources in Western Europe. Medieval chronicles point to abnormal weather conditions in Summer 1258, with economic records highlighting delayed and poor harvests which likely aggravated ongoing grain shortages. Stothers (2000) suggested that frequently cold and rainy weather lead to widespread crop damage and famine, also noting the outbreak of plague across Europe and the Middle East in 1258-59. Suggestions that the Samalas eruption can be linked to the initiation of the "Big Bang" diversification event which led to the Branch 1 strain of *Yersinia pestis* responsible for the Black Death in Europe (Fell et al., 2020) have recently been refuted, with consensus forming instead that this plague proliferation event can be traced to the Tian Shan region much earlier in the 13th century (Green, 2020, Green, 2022). Nonetheless, connections have still been drawn between the anomalous climatic conditions following the eruption and the fall of Bagdad to the Mongol empire in 1258, as well as the subsequent defeat of the Mongol Army at the battle of Ayn Jālūt in 1260 which marked the collapse of the Mongol westward advance (Green, 2020, Di Cosmo et al., 2021). Without a comprehensive understanding of the extent and chronology of climate response to the eruption on a regional-scale, the robustness of these inferred connections between post-eruption climate response and historical events remains difficult to constrain.

The magnitude and spatial distribution of the post-eruption climate response following large volcanic eruptions in known to show a strong seasonal dependency (Stevenson et al., 2017, Toohey et al., 2011). Asymmetric cooling between hemispheres occurs due to seasonal variation in Brewer-Dobson circulation which modulates hemispheric aerosol distribution (e.g Toohey et al., 2011). Asymmetric aerosol distribution combined with enhanced land-albedo feedbacks during NH winter can therefore increase the magnitude of temperature anomalies between hemispheres (Stevenson et al., 2017). Hemispheric temperature contrasts can subsequently drive latitudinal shifts in the Inter-Tropical Convergence Zone (ITCZ), an equatorial band of enhanced rainfall and lower pressure, away from the hemisphere of greatest cooling resulting in hydroclimate perturbations (Broccoli et al., 2006). Therefore, depending on if the Mt. Samalas eruption occurred in Summer 1257 or in early 1258, differences in the magnitude and spatial distribution of resulting SAT and precipitation anomalies are expected.

In this study we utilize a multi-proxy to model comparison to place more precise constraints on the year and season of the Mt. Samalas eruption, tested across the whole window of proposed eruption dates. We utilize both model and proxy constraints to assess regionally heterogeneous impacts of the Mt. Samalas eruption, with reference to proposed historical consequences. Van Dijk et al., 2023 demonstrated the effectiveness of this model multi-proxy approach in their investigation of the regional climatic and social consequences in Scandinavia following the 536/540 CE double eruption event, including incorporating additional archaeological evidence. UK Earth System Model (UKESM1) simulations were run across January and July eruption scenarios and a globally-resolved database of proxy records was collated consisting of tree-ring chronologies, historical sources, and ice core records (Supplementary Sheet 1). Our study demonstrates the ability of a multi-proxy to model

comparison to more precisely constrain the date of the Mt. Samalas eruption, where previous studies have tended to utilize
only a single-proxy approach (Stothers, 2000, Bauch, 2019, Büntgen et al., 2022, Stoffel et al., 2015). The multi-proxy to
model comparison employed in this study is shown to have significant potential in constraining similar uncertainties in eruption
source parameters for other historical eruptions where sufficient coincident proxy records are available.

## 2 Methods

### 2.1 Model Simulations Using the UK Earth System Model (UKESM)

The state-of-the-art interactive aerosol-climate model UKESM1 (Sellar et al., 2019) was used, consisting of the physical global
climate model HadGEM3-GC3.1 with additional configurations for terrestrial and marine biogeochemistry, land and ocean
physics, ocean-sea ice, and dynamic terrestrial vegetation. The model also includes the UK Chemistry and Aerosol (UKCA)
interactive stratospheric-tropospheric chemistry and aerosol schemes (Archibald et al., 2020, Mulcahy et al., 2020). The full
life cycle of stratospheric sulfur and sulfate aerosol particles is included, from injection of $SO_2$, oxidation, particle formation
and subsequent growth, to sedimentation and removal.

The model has a horizontal atmospheric resolution of 1.875° by 1.25° and a 1° by 1° resolution in the ocean, giving a vertical
resolution of 85 levels in the atmosphere and 75 levels in the ocean. This results in well-resolved ocean and atmosphere
dynamics and an internally generated Quasi-Biennial Oscillation (QBO). Coupled ocean-atmosphere simulations were run
with greenhouse gases set to a representative pre-industrial (AD 1850) background state. The difference between a
preindustrial and bespoke 13th century background state is small compared to model internal variability and thus does not
represent a significant limitation of the approach.

Eighteen UKESM eruption-perturbed ensemble simulations were run, with nine simulating a January eruption (JAN1258) and
nine a July eruption (JUL1257), where January and July are winter/summer representatives. Given the preindustrial
background the January/July ensemble groupings are not constrained to a specific year and therefore the two ensemble
groupings have been used to assess the full range of dates proposed for the Mt. Samalas eruption between the years 1256 and
1258. The ensembles sample a range of initial conditions, with the starting phase of both QBO and the El Niño Southern
Oscillation (ENSO) varying between ensembles. For full details of ensemble initial condition classification see Supplementary
Document Table S1. Across the eighteen ensembles only the eruption season and initial conditions were varied, with all other
eruption source parameters held constant and as listed in Table 1.

119 Tg of $SO_2$ was taken from the updated database of VSSI estimates (eVolv2k; Toohey and Sigl, 2017), which is within
error of the 126 +/- 9.6 Tg estimated by Vidal et al. (2016). In our simulations, the injection height is set at 18-20 km to be

consistent with the 1991 Mt. Pinatubo eruption and to allow for lofting of aerosol to higher altitudes in the stratosphere. This height is lower than the estimated 38-40 km column heights (Lavigne et al., 2013, Vidal et al., 2015) however, those column heights refer to the maximum altitude of tephra and ash rather than the height of sulfur injection in the stratosphere or the maximum altitude of the $SO_2$ plume. A 24-hr eruption duration agrees well with Lavigne et al., (2013) who estimated the eruption duration to be 23.8 +/- 10.3 hrs.

An equivalent control ensemble, with identical starting conditions but no eruption perturbation, was run for each of the individual eruption-perturbed ensemble simulations. Anomalies were calculated with respect to a climatological background constructed from the control ensemble mean.

**Table 1: Eruption Source Parameter values used in the ensemble simulations.**

| Eruption Source Parameter | Ensemble Value |
|---|---|
| Volcanic stratospheric sulfur injection (VSSI) | 119 Tg of $SO_2$ |
| Injection Height | 18-20 km |
| Duration | 24hrs |
| Latitude | 8°S (Single grid box) |

**2.2 Multi-Proxy Database of Surface Temperature Impacts**

The proxy database (See Supplementary Sheets 1 and 2 for full details) was constructed by compiling records indicative of changes to climate (tree-ring chronologies, ice core records, and historical sources) that span the range of dates proposed for the eruption with a minimum of annual resolution. Whilst the aim was to create a global database of proxy records, this was significantly impeded by the limited global distribution of all types of proxy records, which show a strong bias to the NH, European, and North American localities. No suitable data was found for this study from Africa or South America.

**2.2.1 Tree-Ring Chronologies**

We include 24 region-specific tree-ring studies, distributed predominantly across the NH, although two studies from Australia and New Zealand are included. The parameter used for temperature reconstruction varies between studies, as does the tree species analyzed and season of reconstruction depending on study location (see Supplementary Sheet 1 for details). SAT anomalies for 1258 and 1259 were taken either directly from the referenced study or calculated from reconstructed SAT

anomalies using a background climatology which was constructed from the 10-year average prior to the eruption. Where studies reported that frost rings were present, they have also been included in the database.

In addition to the 24 region-specific studies, four NH spatially averaged reconstructions have also been incorporated (Wilson et al., 2016, Schneider et al., 2015, Büntgen et al,. 2021, Guillet et al., 2017), along with the N-TREND reconstruction which is spatially resolved for the NH (Anchukaitis et al., 2017, see figure S3 for the spatial distribution of records used in the N-TREND reconstruction). For consistency in this study all tree-ring-reconstructed SAT anomalies are calculated from the 10-year average prior to 1257.

### 2.2.2 Ice Core Records

We include six $\delta^{18}O$ isotope series ice core records from Greenland and Antarctica, where records were chosen on the basis of both annual resolution and an age dating precision of +/- 1 year. Linear regression analysis was applied to calibrate the series to JJA gridded temperature anomalies (with respect to 1990-1960) from the BEST dataset (Rodhe et al., 2020). An additional SAT constraint is also included in Greenland for Summer 1258 from analysis by Guillet et al., (2017) who utilised three Greenland ice cores at GRIP, CRETE, and DYE3 (Vinther et al., 2010) to calculate a clustered SAT anomaly for the region. Additional ice core records were investigated to expand this analysis such as the Illimani Ice Core in Bolivia and the Belukha Ice Core in Altai, Siberia; however, these records lacked the annual resolution required to constrain abrupt temperature changes associated with volcanic eruptions and/or the age dating precision to clearly identify signatures from the Samalas eruption.

### 2.2.3 Historical Sources

Historical sources consist predominantly of medieval chronicles which refer to abnormal and/or extreme weather events in the years 1258-59. Analysis of medieval chronicles and economic records for the years 1258-59 by Guillet et al., (2017) form the basis of historical constraints in Europe. Additional chronicles include references from the Russian Annals in the Altai Mountains (Borisenkov and Pasetsky, 1988, Guillet et al., 2017), the Chronicle of Novgorod from Central Russia (Stothers, 2000), the Þorgils Saga Skarða in Iceland (Bierstedt, 2019), and Azuma Kagami from Japan (Farris, 2006). References to abnormally dark lunar eclipses are also included from the Chronicle of the Abbey of St. Edmunds (Stothers, 2000) and the Annales Ianuenses (Guillet et al., 2017). Additional historical sources across Europe, as well as some from the Middle East, report plague, famine, and economic crises for 1258-60 (Stothers, 2000). These sources have not been incorporated into the database as they refer to social and economic disturbances rather than making direct references to abnormal climatological phenomena.

### 2.3 Simulated SAT Anomalies

### 2.3.1 Northern Hemisphere Average

Spatially averaged NH summer land SAT anomalies were calculated for the mean of the nine JUL1257 and JAN1258 ensembles respectively. Constraints were applied to model-simulated surface temperature outputs to make them most comparable to tree-ring reconstructions (Latitudes of 40°N-75°N, land only, June-July-August (JJA) was taken as representative of the growth season.) Using four NH spatially averaged tree-ring chronologies (Wilson et al., 2016, Schneider et al., 2015, Büntgen et al., 2021, Guillet et al., 2017 – see Supplementary Document, Figure S1 for individual chronology comparison) a mean NH summer SAT anomaly time series was also calculated to which the model-simulated anomalies are compared.

### 2.3.2 Spatially Resolved Comparison

Spatially resolved model-simulated summer (JJA) SAT anomalies were calculated globally and for the NH between 1258-1259. This time window was chosen because the JUL1257 and JAN1258 ensemble simulations show the greatest divergence in SAT anomalies for Summers 1258-59. Model-simulated NH SAT anomalies were re-gridded and masked to facilitate more direct comparison with the N-TREND dataset (Anchukaitis et al., 2017). An analysis of variance (ANOVA) test was performed by eruption season to determine at which grid points the variance in means between the 1258 and 1259 perturbed ensembles exceeded 95% significance relative to the control ensemble simulations, where the null hypothesis was that there was no difference between the mean grid point anomalies for the perturbed 1258 and 1259 ensembles compared to the control ensemble.

### 3. Results

### 3.1 Multi-Proxy to Model Comparison

### 3.1.1 Northern Hemisphere

Figure 1 shows that the JUL1257 ensemble mean (solid blue line) is the only eruption scenario to lie consistently within 2σ of the tree-ring mean (grey band around the black line), with good agreement with tree-ring-reconstructed anomalies for both the timing and magnitude of peak cooling across the whole period (1257-1262). The JAN1258 eruption ensemble mean (solid pink line) also results in peak cooling occurring in Summer 1258 although the magnitude of model-simulated cooling is much greater (by over 1°C) than the peak tree-ring reconstructed cooling. Across individual JAN1258 eruption ensembles (shown in Supplementary Document, Figure S2) only two lie within 2σ of the tree-ring mean for the whole period. By contrast for the individual JUL1257 eruption ensembles seven lie within 2σ of the tree-ring mean for the whole period (Supplementary Document, Figure S2). Both JUL1256 (dashed blue line) and JAN1257 (dashed pink line) eruption scenarios result in peak

cooling occurring a year early relative to the tree-ring-reconstructed mean and across both scenarios no individual ensemble members lie within 2σ of the tree-ring mean for 1257-1260.

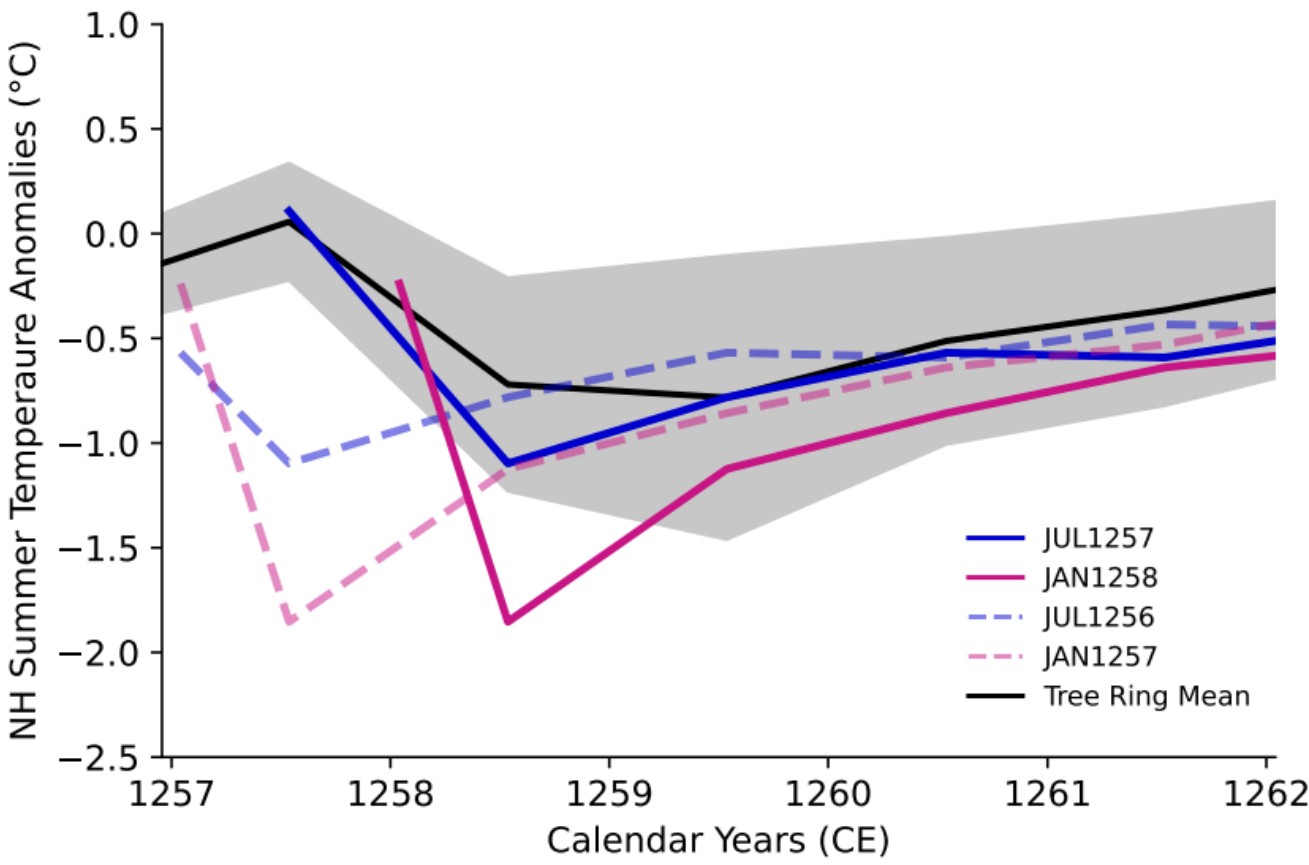

**Figure 1. NH Summer June-July-August surface temperature anomalies. Blue: JUL1257 ensemble mean. Pink: JAN1258 ensemble mean. Solid and dashed lines indicate different eruption years. Black line shows the mean of the tree-ring-reconstructed summer surface air temperature anomalies and grey band shows 2σ around the tree-ring mean. Tree-Ring data: Wilson et al., (2016), Schneider et al., (2015), Büntgen et al., (2021), Guillet et al., (2017). See Supplementary Sheet 2 for details of each tree-ring study.**

When compared to the N-TREND spatially resolved tree-ring reconstructed anomalies (first column in Figure 2) the JUL1257 ensemble mean (third column in Figure 2) results in more consistent agreement with the magnitude and spatial distribution of SAT anomalies across the NH for Summers 1258 and 1259. For Summer 1258 the mean grid point difference between N-TREND reconstructions and the JUL1257 Ensemble mean is +0.19 (σ = 1.08) whilst the difference with the JAN1258 Ensemble mean is +0.78 (σ = 0.99), with the JAN1258 ensemble mean (second column in Figure 2) tending to overpredict summer SAT anomalies relative to N-TREND reconstructions. Across the US West Coast and Central and Northern Europe

N-TREND reconstructions suggest cooling of -1°C to -2°C whereas model-simulated anomalies for the JAN1258 ensemble
mean are on the order of -2°C to -3°C. By contrast the JUL1257 ensemble mean shows widespread but more moderate negative
SAT anomalies of -1°C to -2°C across the NH for Summer 1258 and thus achieves better agreement with N-TREND
reconstructions. Nonetheless, model-simulated anomalies for the JUL1257 mean suggest cooling of up to -2°C in Central and
Northwest Asia which is an underprediction relative to N-TREND tree-ring reconstructions which suggest cooling of up to -
3°C in Summer 1258.

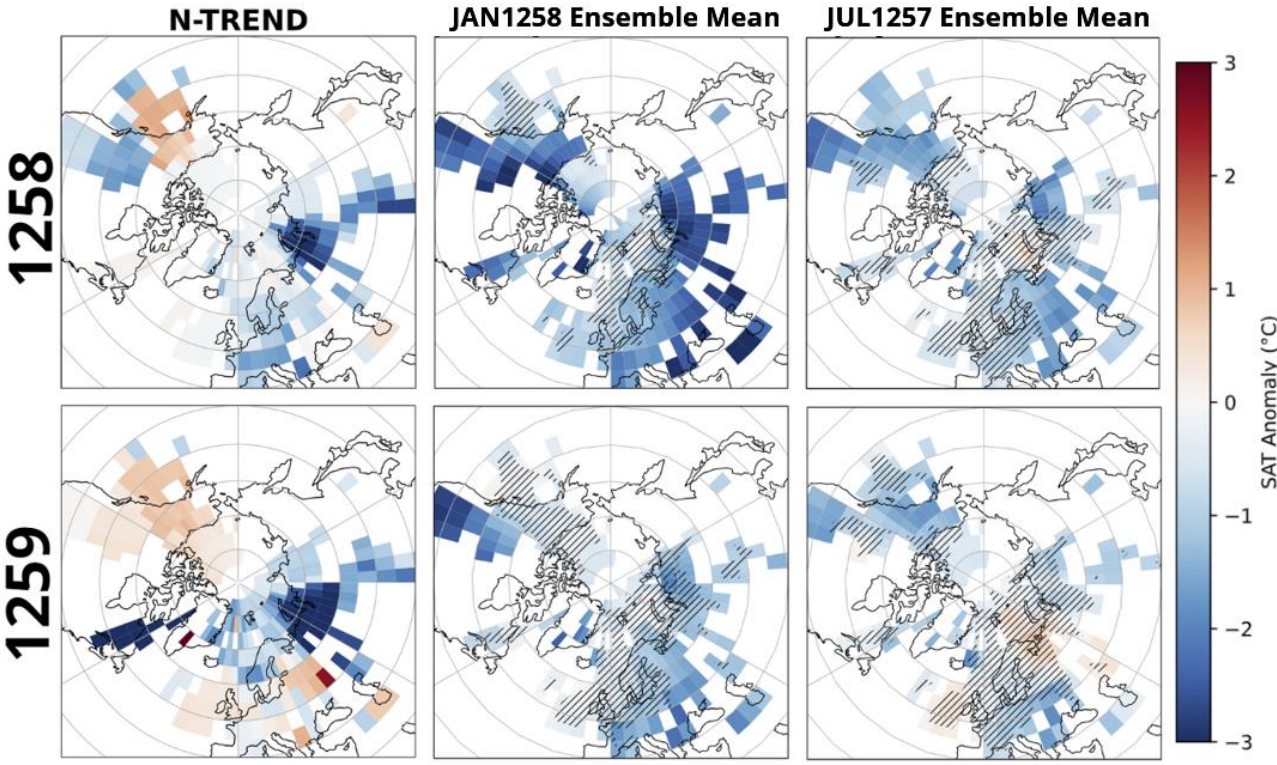


**Figure 2. Spatially resolved N-TREND-Model comparison for June-July-August average 1258-59 for the Northern**
**Hemisphere (40-90°N). N-TREND data from Anchukaitis et al., (2017). First column shows N-TREND reconstructed**
**summer surface temperature anomalies. Second column shows model-simulated summer surface temperature**
**anomalies for the JAN1258 ensemble mean and third column shows the same for the JUL1257 ensemble mean. Hashed**
**areas show regions of less than 95% significance as determined using a grid box ANOVA analysis.**

Greater regional variability is seen in N-TREND SAT anomalies for summer 1259. N-TREND reconstructions suggest
negative SAT anomalies in Northern Eurasia and Quebec of up to -3°C and between -1°C to -2°C in Northern Europe and
Central Asia. Positive SAT anomalies of up to +1°C are seen in Alaska and Western Europe. The JAN1258 ensemble mean

shows continued widespread negative anomalies of -1°C to -3°C across the whole NH and thus does not achieve consistent agreement with reconstructed anomalies across North and Western Europe, US West Coast or Alaska. By contrast, the JUL1257 ensemble mean shows moderate positive SAT anomalies in Northern and Western Europe of up to +0.5°C although still somewhat under predicts the magnitude of cooling in Central and Northern Asia and the US East coast relative to N-TREND reconstructions.

Notably neither the JUL1257 or JAN1258 ensemble mean achieves agreement with positive SAT anomalies reconstructed in Alaska for Summers 1258 and 1259. Across individual ensembles (See Supplementary Document, Figures S4-S7) only 1 JUL1257 and 4 JAN1258 ensembles show positive SAT anomalies in this region. Of these ensembles three are classified as having warm phase ENSO initial conditions. N-TREND reconstructions show moderate SAT anomalies in Alaska from 1255-56, with strong positive SAT anomalies first appearing in reconstructions for Summer 1257 (shown in Supplementary Document, Figure S8).

### 3.1.2 Globally-Resolved Multi-Proxy Constraints

Model-simulated SAT anomalies for a JUL1257 eruption (first row) and a JAN1258 eruption (second row) across Summers (JJA) 1258 (left) and 1259 (right) are shown in Figure 3 with symbols denoting the degree of agreement wit multi-proxy-reconstructed SAT anomalies. The locations and SAT anomalies constrained by proxy records are shown in Figure S3. Overall, the JUL1257 ensemble mean shows more consistent agreement with proxy SAT constraints across Europe, Asia, and North America, whilst the JAN1258 ensemble mean tends to overpredict the magnitude of negative SAT anomalies relative to quantitative proxy constraints.

Summer 1258

For Summer 1258 large negative SAT anomalies are well constrained across Central Asia, with tree-ring reconstructions suggesting cooling in the region of up to -0.4°C in Tibet and -1.1°C in Mongolia (Xu et al., 2019, Davi et al., 2015, Davi et al., 2021) as well as the presence of frost rings late in the 1258 growth season (D'Arrigo et al., 2001). In Japan the "Mirror of the East" historical chronicle refers to persistent cold and wet conditions. Negative SAT anomalies are constrained across northern Russia, with cooling up -2.7°C (Briffa et al., 2013), and frost rings also present in the Polar Urals, Siberia (Hantemirov et al., 2004). Both JAN1258 and JUL1257 model ensembles suggest strong cooling across Central and Northern Asia, however, only the JUL1257 ensemble mean lies consistently within +/-1°C of the proxy constraints whilst anomalies of up to -4°C for the JAN1258 ensemble mean are an overprediction relative to proxy constraints (see filled symbols on Figures 3C and 3E).

Across Northern, Central, and Western Europe negative SAT anomalies are constrained by a combination of tree-ring reconstructions, with moderate cooling of up to -0.3°C in Europe (Büntgen et al., 2011), and a multitude of medieval chronicles across France, Germany, and England which refer to cold and wet conditions (Guillet et al., 2017). The JUL1257 ensemble

shows SAT anomalies of up to -1°C in good agreement with proxy constraints on the magnitude of cooling (see filled triangles
across Europe in Figure 3A), whereas the JAN1258 ensemble mean shows cooling between -2°C and -3°C. The Þorgils Saga
Skarða in Iceland also refers to abnormally cold and wet weather during 1258 (Bierstedt, 2019) although ice core
reconstructions in Greenland show only minor SAT anomalies of between + 0.09°C and -0.1°C (Guillet et al., 2017, Vinther
et al., 2010, Fischer et al., 1998, Vinther et al., 2006). JAN1258 and JUL1257 ensemble means suggest coolings of up to -3°C
and -2°C respectively and so both overpredict the magnitude of cooling across Greenland. Ice core records across Antarctica
all show moderate negative SAT anomalies of up to -0.1°C. The JUL1257 ensemble mean does show negative anomalies
across three of the four ice core sites, although the magnitude of model simulated cooling (-2°C to -3°C) is much greater. The
JAN 1258 ensemble shows more variable SAT anomalies with both warming and cooling anomalies across Antarctica.

SAT anomalies are more variable across North America with reconstructions in Eastern Canada showing both positive and
negative SAT anomalies of 0.09°C and -1.6°C respectively (Gennaretti et al., 2014, Moore et al., 2001). The JUL1257
ensemble shows agreement with both proxy constraints suggesting cooling of up to -2°C on Baffin Island and showing no
significant SAT anomaly in the Quebec region, whilst the JAN1258 ensemble mean shows cooling of up to -3°C and thus
overpredicts cooling in both regions. Tree-ring reconstructions in the Western US and Canadian Rockies suggest negative SAT
anomalies of up to -0.5°C and -1.6°C (Martin et al., 2020, Luckman et al., 2005). The JAN1258 ensemble overpredicts the
magnitude of cooling relative to proxy constraints with SAT anomalies of up to -3°C whilst the JUL1257 ensemble mean
shows more moderate anomalies of up to -2°C. Positive SAT anomalies are well constrained by tree-ring reconstructions in
the Gulf of Alaska with warming of up to 0.1°C and 0.9°C (Wiles et al., 2014), however, neither JAN1258 nor JUL1257
ensemble means show positive SAT anomalies in this region.


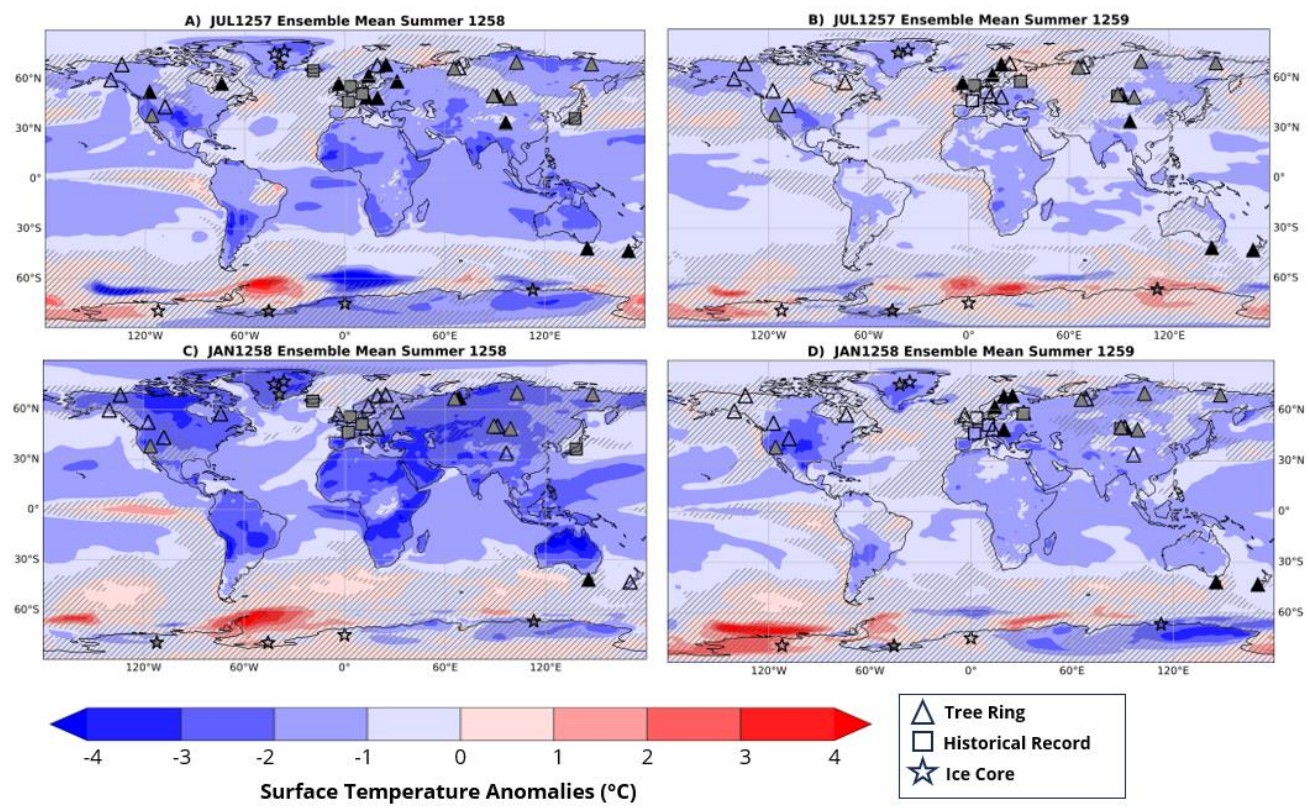

**Figure 3: Globally-resolved multi-proxy-model comparison visualized for summers (JJA) 1258 and 1259. Symbols denote proxy data type and red/blue shading shows model-simulated surface air temperature anomalies for JUL1257 eruptions (a-b) and JAN1258 eruptions (c-d) ensemble means. Surface air temperature anomalies were calculated relative to a 10-year background climatology constructed from the control ensemble mean. Hashed lines denote anomalies at <95% significance as determined by a grid point ANOVA analysis. Black filled symbols denote agreement within +/- 1°C between model-simulated anomalies and quantitative proxy records. Grey filled symbols denote qualitative agreement with proxy records. Locations and proxy-constrained SAT anomalies are shown in Figure S3.**

Summer 1259

Persistent strong negative SAT anomalies are well constrained for Central Asia with tree-ring reconstructions suggesting continued cooling of up to -2.3°C in Mongolia and -0.3°C in Tibet (Davi et al., 2015, Xu et al., 2019) as well as frost rings present early in the 1259 growth season (Churakova et al., 2019, D'Arrigo et al., 2001). The Russian Chronicle of Novgorod refers to abnormal summer snowfall in the Altai mountains and unusual summer frost days (Stothers, 2000, Borisenkov and Pasetsky, 1988) with tree-ring reconstructions showing continued negative SAT anomalies across Northern Russia of up to -4°C (Briffa et al., 2013). Both JAN1258 and JUL1257 ensemble means show moderate negative anomalies of between -1°C

and -2°C in Central Asia, however, only the JAN1258 ensemble mean shows stronger negative SAT anomalies of up to -3°C
in Northern Russia whilst anomalies for the JUL1257 ensemble mean do not exceed 95% significance. Tree-ring
reconstructions show some positive SAT anomalies across Central Europe of up to 0.2°C (Büntgen et al., 2011) and in Western
Europe medieval chronicles refer to a hot and dry summer (Guillet et al., 2017). Very moderate SAT anomalies across Europe
are only shown by the JUL1257 ensemble mean, whilst the JAN1258 ensemble continues to show cooling in the region of up
to -2°C. Ice core records in Greenland show very moderate SAT anomalies of 0.02°C (Fischer et al., 1998) and -0.009°C
(Vinther et al., 2006) however, both JAN1258 and JUL1257 ensemble means show cooling across Greenland of -1°C to -2°C.

Negative SAT anomalies persist across Eastern North America with cooling of up to -2.6°C (Gennaretti et al., 2014). Cooling
of up to -2°C is shown by both ensemble means across Baffin Island (NE Canada), however, neither ensemble achieves
agreement with the stronger cooling signal in Quebec. Along the Western coast of North America tree-ring reconstructions
yield positive SAT anomalies of up to +0.8°C in the Missouri River Basin and +0.3°C in the Canadian Rockies (Martin et al.,
2020, Luckman et al., 2005). Both ensembles continue to show cooling along the US West Coast although the magnitude is
more moderate for the JUL1257 ensemble. In the Gulf of Alaska tree-ring reconstructions suggest continued positive SAT
anomalies of up to +0.7°C (Wiles et al., 2014). The JAN1258 ensemble mean shows no anomalies exceeding 95% significance
in Alaska whilst the JUL1257 ensemble mean shows continued cooling of up to -2°C and so neither eruption scenario achieves
good agreement with proxy constraints. Ice core records across Antarctica continue to show moderate cooling anomalies of up
to -0.15°C apart from the Law Dome (Plummer et al., 2012 - 112°E) core which shows a moderate warming anomaly of +0.16.
The JUL1257 ensemble mean does show a warming anomaly in the east of Antarctica, however cooling anomalies are more
isolated in west Antarctica compared to those reconstructed from ice core records. The JAN1258 ensemble mean shows poor
agreement with ice core records with warm SAT anomalies in the east of Antarctica and cool SAT anomalies in the west.
**3.2 Re-evaluating Evidence for a January 1258 Eruption Date**
Two lines of evidence have previously been invoked to support an earlyy 1258 eruption date: references in medieval chronicles
to a "dark lunar eclipse" in mid-May 1258 (Stothers, 2000) and peak sulfate fall out in Greenland ice cores in early 1259
(Hammer et al., 1980). Figure 4a shows model-simulated Stratospheric Aerosol Optical Depth (SAOD) averaged across
western Europe for the years following the eruption with the black line showing the eVolv2k SAOD for the PMIP4 forcing
which is based on a July 1257 eruption date (Toohey & Sigl, 2017). Marked by the vertical grey lines are the dark lunar eclipses
of 18-19th May 1258 and 12-13th Nov 1258 identified by Stothers (2000) in the Bury Saint Edmunds Abbey chronicle and
Guillet et al., (2017) in the Annales Ianuenses respectively, where for the moon to appear dark an SAOD > 0.1 is needed
(Stothers, 2000). Whilst a JAN1258 eruption scenario does result in a later SAOD peak, both a JUL1257 and a JAN1258
eruption result in SAOD >> 0.1 during both May and November 1258 and therefore either eruption scenario could account for
observations of darkened lunar eclipses. This is supported by findings from Guillet et al., (2023) that a dark total lunar eclipse
is most likely to be observed 3 to 20 months following an explosive eruption with a VSSI exceeding 10 Tg. SAOD in the
UKESM for both JUL1257 and JAN1258 ensembles decays more rapidly than the eVol2k SAOD used for the PMIP4 forcing,
for which SAOD remains > 0.1 until mid-1260 AD..

**Table 2: Comparison between ice core average (from Table S2) and ensemble means for JUL1257 and**
**JAN1258 model simulations for the timing of SO4 rise, peak, and total deposition in Greenland and**
**Antarctica.**

| | JUL1257 Ensemble Mean | JAN1258 Ensemble Mean | Ice Core Mean (Average from Table S2) |
|---|---|---|---|
| SO$_4$ Rise Greenland | Jan 1258 | May 1258 | February 1258 |
| SO$_4$ Rise Antarctica | Oct 1257 | March 1258 | August 1257 |
| **Offset in SO$_4$ Rise (months, Antarctica - Greenland)** | **3** | **2** | **6** |
| SO$_4$ Peak Greenland | July 1258 | Jan 1259 | March 1259 |
| SO$_4$ Peak Antarctica | February 1258 | Sept 1258 | July 1258 |
| **Offset in SO$_4$ Peak (months, Antarctica - Greenland)** | **5** | **4** | **8** |
| Greenland Sulfate Deposition (Kg km-2) (Toohey & Sigl 2017) | **26.8** | **23.2** | 105 |
| Antarctica Sulfate Deposition (Kg km-2) (Toohey & Sigl 2017) | **54.3** | **28.0** | 73 |
| **Greenland/Antarctica** | **0.49** | **0.83** | **1.4** |



Figures 4B and 4C show ice sheet-averaged model-simulated sulfate deposition across Greenland and Antarctic ice sheets
respectively, for JUL1257 and JAN1258 eruption scenarios. The mean timing of the onset of sulfate rise and the timing of
peak sulfate deposition across four high resolution ice cores (see Table S2) are also shown as vertical black lines. Table 2
shows a comparison between ice core average (from Table S2) and ensemble means for JUL1257 and JAN1258 model
simulations for the timing of SO4 rise, peak, and total deposition in Greenland and Antarctica. The JUL1257 ensemble shows

that the onset of sulfate deposition occurs in January 1285 in Greenland and October 1257 in Antarctica, with peak deposition occurring in July 1258 and February 1258 respectively. By contrast, the JAN1258 ensemble shows sulfate rise beginning in May and March 1258 in Greenland and Antarctica respectively, with peak deposition in Jan 1259 and September 1258. Across four high resolution ice cores (Table S2) the mean timing of the onset of sulfate deposition occurs in February 1258 in Greenland and August 1257 in Antarctica, thus showing closest agreement with the timing of the simulated onset in sulfate deposition for a July 1257 eruption date, where only an eruption in summer 1257 can account for the beginning of sulfate deposition in Antarctica in autumn 1257. Across the four ice core records mean peak deposition in Greenland occurs in March 1259 whilst mean peak deposition in Antarctica occurs earlier in July 1258. The timing of peak deposition therefore shows better agreement with simulated peak deposition for a January 1258 eruption date with peak model-simulated deposition occurring too early in the JUL1257 ensemble relative to ice core records. Ice core records also suggest an asymmetry in sulfate dispersal and deposition, with the onset of sulfate deposition and peak deposition in Antarctica being 6 and 8 months ahead of Greenalnd respectively. The JUL1257 ensemble shows a greater degree of asymmetry compared to the JAN1258 ensemble, although offsets of 3 and 5 months between Antarctica and Greenland are still lower than the offset suggested by the ice core record. Whilst the offset between the onset in sulfate deposition and peak sulfate deposition is approximately a year in both Greenland (13 months) and Antarctica (11 months), the offset for model-simulated deposition in the UKESM is considerably shorter across both JUL1257 and JAN1258 ensembles (ranging from 4-8 months). This may represent a limitation specific to the UKESM, for example with too weak poleward transport of volcanic aerosol (or midlatitude deposition too strong). This weaker poleward transport may also contribute to the lower magnitude of total polar deposition simulated by the UKESM relative to ice core records, also seen in an earlier version of the UK climate model for the 1815 eruption of Mt. Tambora (Marshall et al., 2018). Both model-simulated ensembles show greater total sulfate deposition in Antarctica compared to Greenland, whilst ice core records suggest the opposite asymmetry favouring greater deposition in Greenland.

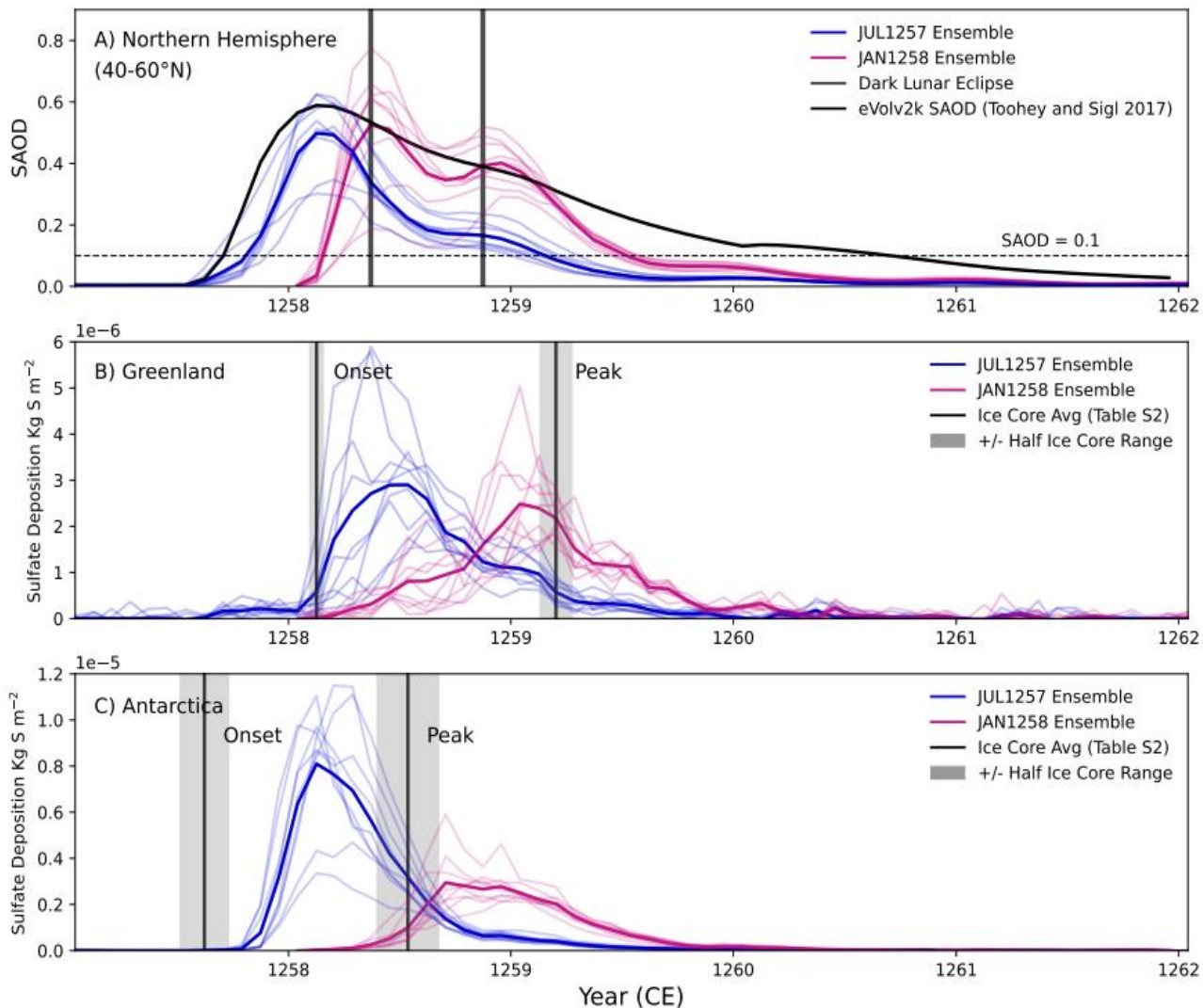

383

**Figure 4: A) Model-simulated Stratospheric Aerosol Optical Depth (SAOD) timeseries averaged across Western**
**Europe (Lat: 40-60°N, Longitude: 10W-10°E) for JUL1257 (blue line) and JAN1258 (pink line) eruption scenarios**
**where bold lines are the ensemble means. Vertical grey bars denote historical records of dark lunar eclipses in May**
**1258 (England) and November 1258 (Genoa). The dashed horizontal line at SAOD = 0.1 denotes the minimum SAOD**
**required for a dark lunar eclipse. B-C) Model-simulated sulfate deposition (kg S m-2) over Greenland and Antarctica**
**ice sheets for JUL1257 (Blue) and JAN1258 (Pink) eruption scenarios. The region of deposition is limited to the single**
**model grid box containing NEEM (Greenland) and WDC (Antarctica) ice core drill sites respectively. Black lines are**
**timeseries from NEEM and WDC ice core records respectively (Sigl et al., 2015) with grey bars showing +/- 1-year**
**uncertainty for the timing of peak deposition.**

## 4. Discussion

### 4.1 Constraining the Eruption Year and Season of the Mt. Samalas eruption

As shown in Figure 1, both JUL1256 and JAN1257 eruption scenarios result in peak surface cooling occurring a year too early relative to tree-ring reconstructed SAT anomalies. Thus, an early eruption date such as Spring 1256, as proposed by Bauch (2019), is, based on our analysis, unfeasible for the Mt. Samalas eruption. Across the two remaining eruption scenarios only the JUL1257 eruption ensemble lies consistently within 2σ of the tree ring-reconstructed mean between 1258-1262. By contrast, the JAN1258 eruption ensemble results in peak cooling being over 1°C greater than tree ring reconstructions for Summer 1258.

When compared with spatially resolved N-TREND reconstructions model-simulated SAT anomalies for the JAN1258 ensemble mean continue to consistently overpredict the magnitude of cooling for both 1258 and 1259 across the NH (Figure 2). Whilst the spatial agreement of SAT anomalies for the JUL1257 ensemble mean does not precisely replicate those reconstructed from the N-TREND tree ring record it does show widespread but less extreme negative SAT anomalies across the NH and thus achieves better agreement with reconstructed anomalies on a regional basis. This is supported by comparison between globally-resolved model-simulated ensemble mean SAT anomalies and multi-proxy-reconstructed anomalies where the JAN1258 ensemble mean consistently over predicts the magnitude of cooling in key regions across Europe, Central Asia, and the US, whilst the JUL1257 ensemble mean results in generally better agreement across these regions with more moderate negative SAT anomalies.

A re-evaluation of evidence for a January 1258 eruption finds that a Summer 1257 eruption can also satisfy constraints from both ice core records and historically documented dark lunar eclipses. Model-simulated ice sheet sulfate deposition shows that for eruption scenarios 6 months apart there is a distinguishable off-set in the timing and magnitude of deposition, with the bi-polar Greenland/Antarctica ratio of simulated sulfate depositions between northern and southern hemisphere ice sheets varying depending on eruption season. The onset of model-simulated sulfate deposition in Greenland and Antarctica for the JUL1257 ensemble mean achieves better agreement with ice core record tie points, where the onset of sulfate deposition in Antarctica in August 1257 can only be aligned with an eruption during or prior to Summer 1257. Nonetheless, the timing of peak model-simulated sulfate deposition achieves better agreement with the JAN1258 ensemble mean, potentially reflecting limitations in aerosol transport within the UKESM leading simulated aerosol decay being too-rapid (Marshall et al., 2018).

Overall, the multi-proxy to model comparison utilized in this study provides a clear distinction between JULY1257 and JAN1258 eruption scenarios, with better agreement between proxy reconstructions and model-simulated anomalies being shown for a July 1257 eruption date. This is consistent with the May-August 1257 date constraint suggested by Guillet et al., (2023) based on their analysis of contemporary reports of total lunar eclipses, combined with tree ring-based climate proxies

and aerosol model simulations. Although consensus has converged on a Summer 1257 date for the Samalas eruption, it remains
to be seen if a more precise constraint (i.e. to a specific month) could be achieved given current model and proxy uncertainties
(as discussed in Section 4.3 below). Nonetheless, this four-month window remains an improvement upon previous dating
uncertainty and is still sufficient at present for interrogating the both climatic and human consequences following the eruption.
**4.2 Regionally Heterogenous Climate Response**
Multi-Proxy SAT reconstructions highlight the regionally heterogenous climate response following the Mt. Samalas eruption.
The largest negative SAT anomalies occur across Central Asia and Northern Russia, with cooling of -2°C to -4°C between
1258 and 1259, making these as some of the most severely impacted regions in the NH. The role of sudden and severe cooling
associated with the Mt. Samalas eruption in the collapse of the Mongol westward advance is therefore plausible (Di Cosmo et
al., 2021). Alongside references to extreme and abnormal weather conditions, the Azuma Kagami in Japan also highlights the
severity of the Shôga famine between 1257-60 (Farris, 2006). Model-simulated cooling of up to -1°C across Japan suggest the
severity of this famine could have been amplified by the climate response to the Mt. Samalas eruption in 1258-59. Similar
evidence exists in the Middle East with famine and pestilence reported across Syria, Iraq, and Southern Turkey in 1258
(Stothers, 2000). A model-simulated JUL1257 eruption scenario suggests cooling of up to -2°C in the region and thus supports
a possible association with the Mt. Samalas eruption. Reconstructed and model-simulated SAT anomalies suggest less severe
cooling across Europe for 1258 (<-1°C) and relative warming in the region for 1259. Nonetheless, this is still associated with
significant economic and social disturbances, with historical records reporting famine and social unrest (Guillet et al., 2017,
Stothers, 2000). Van Dijk et al., (2023) found similar regional variability in both the climatic effects and social consequences
across Scandinavia following the 536/540 double eruption event, finding the severity of impacts depended heavily both on
local topography and the subsistence methods employed by different communities.
Büntgen et al., (2022) found negative TRW growth anomalies along the US West Coast for Summer 1257 and suggest this is
evidence for the onset of climate perturbations in the NH before the end of the 1257 growth season and therefore support a
Summer 1257 eruption date. Model-simulated anomalies for a JUL1257 eruption, however, show no significant anomalies
occurring across the US or NH during Summer 1257 (see Supplementary Document, Figure S9). The presence of frost rings
in 1257 and 1259 (Salzer and Hughes, 2007), but not in 1258, also contradicts model-simulated cooling, which consistently
shows the strongest cooling in 1258. Model-simulated SAT anomalies for a JUL1257 eruption only show significant negative
SAT anomalies in late summer 1257 across South America, Africa, and Oceania, with cooling in some regions of up to -2°C,
although without well-dated, geographically distributed climate proxies in the SH it will remain difficult to resolve whether
this is potential model/proxy discrepancy in the SH (Neukom et al., 2014). However, if extreme and sudden cooling did occur
in the SH would be expected to have significant, but as yet unknown, consequences for communities and civilizations in the
Southern Hemisphere. Additional large volcanic eruptions in 1269, 1276 (UE5), and 1286 (UE6), which combined with the

eruption in 1257, make the sulfate loading in the 13th century two to ten times larger than any other century in the last 1500 years (Gao et al., 2008, Guillet et al., 2023), and may have led to further climate anomalies with effects on impacted civilisations being prolonged throughout the latter half of the 13$^{th}$ century. For example, the first settlement of New Zealand most likely occurred between 1250–1275, with suggestions this may have reflected a climate-induced migration associated at least in part with the impacts of the Mt. Samalas, and subsequent, eruptions (Anderson, 2016, Bunbury et al., 2022). Apart from recent analysis of the localized impacts and recovery following the Mt. Samalas eruption (Malawani et al., 2022) the general sparsity of currently available proxy data across the SH precludes definitive conclusions as to climate and social response in the SH following the Mt. Samalas eruption.

Notably, neither the JAN1258 or JUL1257 ensemble mean replicates reconstructed positive SAT anomalies in Alaska for Summer 1258 or the wider US west coast for Summer 1259. Across individual realisations only five show positive SAT anomalies in this region for 1258-59, and of these ensembles 3 are classified as having warm phase ENSO initial conditions. N-TREND spatially resolved reconstructions show positive SAT anomalies over Alaska from Summer 1257-59, with warm phase El Niño-like conditions during this period being supported by positive tree ring-reconstructed temperature anomalies in Alaska (Guillet et al., 2017) and the muted absolute temperature signal seen in $\delta^{18}$O coral reconstructions from the central tropical Pacific (most likely explained by the superposition of a volcanic cooling signal and a warm El Niño-like signal (Dee et al. 2020, Robock 2020)). The role of volcanic eruptions in perturbing ENSO remains debated (Mann et al. 2005, Stevenson et al. 2018, Dee et al. 2020, Robock 2020) although climate simulations and proxy records suggest an increased probability for the occurrence of an El Niño event in the first or second year after a large volcanic eruption (see McGregor et al 2020 for a full review). El Niño-like warm conditions may therefore have prevailed at the time of the Mt. Samalas eruption with subsequent volcano-ENSO interactions potentially acting to enhance these pre-existing El Niño-like conditions.

**4.3 Model and Proxy Limitations**

The majority of temperature anomaly constraints applied in this study are provided by SAT tree ring reconstructions, where tree-ring data has been shown to effectively record extreme cooling events synchronous with evidence for explosive volcanic eruptions over the last two millennia without chronological errors (Stoffel et al., 2015, Büntgen et al., 2020; Büntgen et al., 2016; Sigl et al., 2015) and in good agreement with instrumental measurements following large eruptions (Esper et al., 2013). Lücke et al., (2019) highlight the importance of accounting for biologically based memory effects, which can lead to dampening of volcanic cooling signals in ring width-based chronologies. However, many of the tree ring-based reconstructions utilised in this work do incorporate maximum latewood density (MXD) data as well as TRW (see Figure S3) which has been shown to reduce attenuation of volcanic cooling signals (Esper et al., 2014, Stoffel et al., 2015). Additional temperature anomaly constraints are also provided by historical sources, although these records are inherently more limited being non-quantitative and subjective accounts. The application of historical records to the model-multi proxy framework is limited by the sparsity of historical records, both spatially, being predominantly concentrated in the NH, and temporally, being biased in

frequency, dating accuracy, and traceability towards more recent volcanic eruptions, although have been applied effectively to
understand the aftermath and impacts of the Mt Samalas 1257 eruption (Malawani et al., 2022). In this study ensemble
simulations do show hydroclimate perturbations following the Samalas eruption for both JUL1257 and JAN1258 scenarios
(Figure S10), although due to the limited number of realisations these are not included in our model-proxy framework.
Nonetheless, the magnitude of precipitation anomalies for Summer 1258 being greatest in the equatorial regions, potentially
indicative of a shift in the ITCZ due to asymmetric cooling between hemispheres. Therefore, future studies may also consider
including hydroclimate proxies alongside SAT reconstructions in order to add an additional constraint on model-proxy
agreement.

Model-simulated anomalies are strongly dependent on model set up, including model resolution, modelled stratospheric winds,
aerosol microphysics and sedimentation and deposition schemes (Marshall et al., 2018, Quaglia et al., 2023). In recent model
intercomparison studies (Marshall et al., 2018, Quaglia et al., 2023) UM-UKCA, a previous version of the UKESM, showed
a bias towards stronger transport to the NH extratropics, resulting in a hemispherically asymmetric aerosol load. The spatial
distribution of volcanic forcing can influence subsequent growth of sulfate aerosols and their global distribution, in turn
affecting the persistence of aerosols in the stratosphere (Quaglia et al., 2023). Compared to other global aerosol models UM-
UKCA also has relatively weaker poleward transport, with stronger meridional deposition which may lead to a more
equatorially focussed aerosol distribution and deposition (Marshall et al., 2018). Disentangling large inter-model differences
from the range of model components that contribute to this uncertainty remains challenging, although future multi-model
multi-proxy studies may be of use.

Even within a single model, uncertainties persist in initial eruption conditions (e.g phase of the QBO,) and volcanic forcing
parameters (e.g timing, magnitude, injection height and latitude of eruption). Timmerick et al., (2021) demonstrated that both
the magnitude of forcing as well as its spatial structure can similarly affect proxy–simulation comparisons, particularly in the
NH extratropics. This is supported by Lücke et al., (2023) who demonstrate a significant spread in the temperature response
due to volcanic forcing uncertainties which can strongly affect the agreement with proxy reconstructions. The VSSI estimate
used in our study was taken from the eVolv2k reconstruction (Toohey and Sigl, 2017) and is within error of other $SO_2$ emission
estimates for the Mt Samalas eruption (Vidal et al., 2016, Lavigne et al., 2013). Whilst maximum plume heights have been
estimated for the Samalas eruption (~ 43 km Vidal et al., 2015) the $SO_2$ injection height remains unknown. In our simulations
the injection height is set to 18-20 km, which may be too low, but does allow for lofting of sulfate aerosol.  Moreover, Stoffel
et al., (2015) found that increasing plume height from 22-26 km to 33-36 km for simulations of the Mt Samalas 1257 eruption
increased the magnitude of the peak post-eruption NH JJA temperature anomaly to -4°C for a January eruption and -1°C and
-2°C for May and July eruption scenarios respectively. A plume height greater than the 20km used in our study would therefore
likely further enforce our central conclusion that better agreement between proxy and model-simulated temperature anomalies
is achieved for a summer 1257 eruption date, whilst a greater plume height for a January 1258 eruption would only further

overpredict the magnitude of post-eruption cooling. Timmreck et al., (2009) also highlight the strong dependence of model-simulated post-eruption climate responses following large volcanic eruptions on the aerosol particle size distribution due to self-limiting effects of larger particles (Pinto et al., 1989). These particle characteristics are difficult to constrain retrospectively for historical eruptions such as Mt. Samalas and thus represent a significant uncertainty when simulating historical eruptions.

Model-simulated ice sheet sulfate deposition (Fig. 4) shows a clear distinction between JAN1258 and JUL1257 eruption scenarios in both the timing and hemispheric distribution of deposition. For the same stratospheric $SO_2$ injection, an eruption during NH summer results in more pronounced asymmetric polar deposition, with the magnitude of peak deposition in Antarctica being nearly a factor of 3x greater than for an eruption during the SH summer, likely due to the stronger branch of the Brewer-Dobson circulation and seasonal effects on aerosol transport and depositional processes. Nonetheless, model simulations of the Tambora 1815 eruption by Marshall et al., 2018 found a strong model dependency for the timing, magnitude, and spatial distribution of sulfate deposition. Polar sulfate deposition in the UKESM was half that reconstructed using ice cores for the Tambora 1815 eruption and considerably lower than deposition in the other MAECHAM5-HAM and SOCOL-AER models analysed (See Table S3). In our JUL1257 and JAN1258 simulations for the Samalas 1257 eruption mean total deposition in Antarctica (55-28 Kg km-2) and Greenland (26-23 Kg km-2) was also lower than mean total deposition from ice cores records (73 and 105 Kg km-2 respectively, see Table 2). The overall lower polar total sulfate deposition in the UKESM may be caused by too weak polarward transport (or stronger meridonal deposition) (Marshall et al., 2018). For our JUL1257 and JAN1258 model-simulated sulfate deposition for both eruption scenarios result in Greenland to Antarctica ratios < 1, with a Summer 1257 eruption showing a much more asymmetric distribution (0.49) than a January 1258 eruption (0.83). These ratios however, disagree with the Greenland to Antarctica ratio derived from ice cores (1.4, Toohey and Sigl, 2017) suggesting that, for our Samalas scenarios, sulfate aerosol transport to the NH is too low, thus favouring transport to the SH. Of the four models analysed in Marshall et al., (2018) for the Tambora 1815 eruption, UM-UKCA was one of two models that had greater deposition in Greenland compared to Antarctica (Table S3, Greenland/Antarctica = 1.7). Both the Tambora 1815 and Mt Samalas 1257 were large magnitude eruptions at a similar latitude, therefore this intra-model difference in the asymmetric distribution of sulfate aerosol most likely results from differences in initial conditions used for our simulations (such as the phase of the QBO). Overall, this further highlights the complications of disentangling inter-model differences and intra-model variation due to initial conditions. Given both these inter and intra-model differences in relative hemispheric aerosol distribution and deposition, the current robustness of using hemispheric sulfate deposition ratios to distinguish between eruption scenarios when compared to ice core records is limited, with further work being needed to understand model and starting condition specific effects.

Whilst comparisons of simulated and reconstructed climate responses following large volcanic eruptions have been used routinely and effectively by a multitude of studies (van Dijk et al., 2023, Büntgen et al., 2022, Stoffel et al., 2015 ), there are several limitations to this approach. Given the uncertainties associated with both proxy reconstructions and model simulations neither can be taken as the inherently "correct" baseline with which to fit the other, and thus particular care should be taken

when using model-proxy comparison to validate the correctness of underlying records or model input parameters. For example, the missing tree ring hypothesis (Mann et al., 2012), which has since been widely rejected, proposed that the mismatch between climate simulations and proxy reconstructions resulted from chronological errors due to missing growth rings. This has, however, since been resolved with improved estimates of volcanic forcing (Toohey & Sigl 2017), the inclusion of climate-independent geochronological data, and the greater inclusion of MXD records in tree ring reconstructions (Stoffel et al., 2015; Schneider et al. 2015; Wilson et al., 2016; Anchukaitis et al. 2017). When utilising an array of different proxy data there is the risk of confirmation bias meaning records which show significant agreement with model simulations being given greater weighting than those which show less agreement. A more quantifiable approach to model-proxy comparison, where a greater number of model realisations would allow for more robust statistical evaluation would therefore be a considerable improvement for the application of the model-multi proxy framework. Further to this, better uncertainty quantification for proxy data would enable more robust comparison with model outputs, particularly in multi-model comparison studies.

**5 Conclusions**

We have utilized eighteen aerosol-climate UKESM1 ensemble simulations for the Mt. Samalas eruption in combination with an extensive globally-resolved multi-proxy database for the Mt. Samalas eruption in order to constrain the year and season as well as the regionally heterogeneous climate response following the eruption. Comparison with NH averaged and spatially resolved tree ring reconstructions showed that a Summer 1257 eruption scenario agrees best with reconstructed SAT anomalies, while a January 1258 eruption consistently overpredicts the magnitude of cooling relative to reconstructions. The regionally variable SAT response following the eruption is revealed by multi-proxy reconstructions which lend support to inferred social, economic and historical consequences across Europe and Asia following the eruption. Model-simulated SAT anomalies also suggest the onset of sudden a severe cooling across the SH, with the potential for significant social and economic consequences in impacted communities across South America, Africa, and Oceania. The spatial distribution of SAT anomalies shows sensitivity to initial atmospheric-ocean conditions, with positive SAT anomalies in Alaska being potentially indicative of warm El Niño-like conditions at the time of the eruption, with potential ENSO-Volcano interactions enhancing these conditions further.

Overall, the proxy to model comparison employed in this study has been shown as an effective approach to constrain uncertain eruption source parameters. Model-multi proxy frameworks have similarly been employed by other recent studies (see Guillet et al., 2023 and van Dijk et al., 2023) which have also demonstrated its potential in constraining unknown eruption source parameters as well as regional climatic impacts for historic eruptions where there is sufficient concurrent proxy evidence. A greater global distribution of proxy evidence, especially in the Southern Hemisphere where all types of proxy evidence are sparse, will strengthen this proxy-model framework approach for future analysis. The incorporation of hydroclimate anomalies in particular has the potential to add further independent constraints, although relies on the development of higher resolution records, especially at low latitude sites. Model simulations of polar sulfate deposition also reveal distinct differences in the

timing of ice sheet deposition between the two simulated eruption seasons, although comparison of the magnitude or
asymmetric deposition of sulfate aerosol remains limited by large inter-model differences and complex intra-model
dependencies.

Finally, both proxy-reconstructed and model-simulated surface temperature anomalies highlight the severity of the global
climate response following a large tropical explosive eruption like Mt. Samalas, with historical records confirming widespread
and severe economic and social consequences. This adds further weight to recent calls (Cassidy and Mani, 2022) for increased
global preparedness for the next large magnitude explosive volcanic eruption, given profound global consequences that would
be expected, as clearly demonstrated by the Mt. Samalas eruption.

**Data Availability**
Data has been uploaded to the CEDA archive and is pending review. Catalogue record can be found at
https://catalogue.ceda.ac.uk/uuid/e0221b37aa174dd290c5e105263b59d1 .

**Supplement**
The supplement related to this article is available online at:

**Author Contributions**

LW, LM, and AS jointly conceived the project methodology. LM ran the UKESM model simulations. LW performed the
analysis, visualisation, and writing of the manuscript with supervision from LM and AS. All authors jointly reviewed and
edited the paper.

**Competing Interests**
The authors declare that they have no conflict of interest.

**Acknowledgements**
This research formed part of LWs MSci Thesis at the University of Cambridge. LW would like to thank O. Shorttle and E.
Harper for their additional supervision and encouragement throughout this project, as well as Newnham College for supporting
the presentation of this work through a travel bursary. LM and AS acknowledge support from NERC grant NE/S000887/1
(VOL-CLIM), and AS additionally acknowledges funding from NERC grant NE/S00436X/1 (V-PLUS). The authors would
also like to thank M. Sigl and K. Anchukaitis for providing ice core and tree ring data sets utilized in this project, and N.L
Abraham for technical modelling support. This work used the ARCHER UK National Supercomputing Service
(http://www.archer.ac.uk) and JASMIN, the UK collaborative data analysis facility. We also thank both reviewers for their
insightful and helpful suggestions.

**Financial Support**
This research was supported by NERC grant NE/S000887/1 (VOL-CLIM) and NERC grant NE/S00436X/1 (V-PLUS).

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
