# Peer review of "Utilizing a Multi-Proxy to Model Comparison to Constrain the Season"

_EGUsphere, 2023_

## Referee Comment (RC1)

Review: "Utilizing a Multi-Proxy to Model Comparison to Constrain the Season and Regionally Heterogeneous Impacts of the Mt. Samalas 1257 Eruption" by Laura Wainman et al.

**A: General comments:**

The authors aim to further constrain the date of the mid-13[th] century Samalas eruption in Indonesia, the largest volcanic eruption in the Common Era. For this they use ensembles of aerosol-climate model simulations forced with $SO_2$ injections in either July 1257 or January 1258, the most widely used dates for this eruption. They find that the simulated global climate response for the July 1257 eruption scenario produces better agreement with collated proxy reconstructions for 1257 to 1259 and argue on those grounds that the eruption likely occurred in summer 1257, in agreement with previous papers.

Tropical eruptions often remain difficult to date exactly, because of a lack of documentary records from direct observations and the scarce available information preserved under tropical climates in the areas surrounding the volcanos. The seasonal timing of eruptions can have important effects on the global climate impact. This paper is a welcome contribution to ongoing efforts in this research field. It is well written, with a clear structure and outline of the scientific hypothesis, especially considering this is a master thesis. The setup of the aerosol-climate model experiments appears appropriate (for a non-expert) and I acknowledge that the authors also have collated and included a large set of proxy records from different archives. I would like to invite the authors to address in their revision a number of points (outlined in detail under *Specific Comments*) focusing on three main topics:

1) **The framing:** The previous arguments and references used to argue for a 1258 eruption are somewhat outdated going back to the early 2000s when ice-core records were biannually resolved, large-scale tree-ring records were not fully developed and the source of the eruption had been unknown; in the last years the research community has already largely converged towards accepting a date in 1257, which wasn't really challenged.

2) **The rationale:** A more general discussion would be welcome to which extent we wanted (or should avoid) to constrain external climate forcing through comparisons of simulated and reconstructed climate response. I noted two specific examples below in which the desire for a high model-proxy agreement has led to speculations of dating errors in tree-rings and in ice cores. In both cases these speculations were later rejected based on climate-independent geochronological data (i.e. radiocarbon, tephra). So there is an eminent risk of overfitting climate forcing records by aiming to minimize model-data disagreement. A short discussion on the risks (beside the potential) would be helpful.

3) **The implications.** I have some reservations regarding your interpretation of the ice-core records (outlined in detail below) and also the implications this study has for the use of ice cores to infer volcanic forcing in the past. Existing limitations to constrain the exact timing of tropical eruptions are dominated by the poorly constrained time-lag of sulfate deposition on the ice sheets following a volcanic eruption, and to a less extent by the resolution of the ice-core records. This time-lag is also highly variable among different state-of-the-art aerosol models. I therefore argue that the resulting spread in stratospheric aerosols and the spatio-temporal climate fingerprint will depend foremost on the choice of the aerosol-climate model. Given these model uncertainties I currently do not see a strong case to constrain the climate-independent ice-core estimates of volcanic forcing with one specific aerosol-climate model.

**B: Specific comments:**

**L35:** I doubt that these two references provided the necessary record length, spatial representation and completeness to put the size of the sulfate deposition into a 2500-year context.

**L40:** please replace Wade et al. (2020) with Schneider et al., (2015).

**L45:** I am not aware of anybody citing January 1258 as a potential date, and the only citation in this paragraph (Stothers 2000) speaks of early 1258. Please clarify.

**L51-52:** I find it unlikely that any radiocarbon date on pyroclastic material would provide the necessary precision to contribute to the discussion of the actual calendar year of the eruption. All the more so as the paper describing it is already 20 years old.

**L54:** Besides TRW anomalies the Western US tree-rings (Salzer and Hughes, 2007) also had frost rings in 1257 and 1259 which combined provide strong evidence of a volcanic source for the cooling and which may help to constrain the age of the eruption.

**L57:** The statement "peak sulfate deposition in ice cores is recorded for 1259" lacks a reference and it is also not true for all ice cores. Some ice cores have the peak sulfate levels in January 1259 (NEEM), others in November 2018 (WDC06A). See Table 1 below. For the timing of the eruption the timing of the peak sulfate level is anyhow not of great relevance but rather the timing of the start of volcanic sulfate deposition which is dated between 1257.4 and 1258.2 in four different records from Greenland and Antarctica (Plummer et al., 2012; Sigl et al., 2013).

**L90:** How would spring (April) or fall (September) eruption scenario differ from January and July?

**L92:** I would refrain from calling the database most complete; for your research question you don't need the most complete dataset but the most suited data.

**L140:** Your selection criteria should not be annual resolution but annual age precision. Only tree-rings and documentary records fulfil this requirement and all ice cores in which the Samalas eruption had been securely identified. For these reasons, I would strongly advise you to remove the lake sediment records and the Svalbard record from your analysis all of which contain no clear signatures of Samalas and are not dated to 1258 (±0). On the other hand, many more than the three used ice cores (GRIP, CRETE, DYE3) would be readily available for analysis both from Greenland and from Antarctica.

**L157-159:** A more comprehensive dataset from Greenland would potentially be more skillful than a set of only three ice cores. The PAGES2k Consortium (2017) database contains at least 9 ice cores from both Greenland and Antarctica with annual resolution encompassing the age of Samalas (PAGES2k Consortium 2017).

**L159-169:** As stated above I don't think that any of these climate information provided for these records can with certainty be linked to the years 1257-59.

**L276:** Please cite the original reference (Vinther et al., 2010).

**Figure 3:** What does the absence of symbol indicate (e.g. Japan and Greenland in 1259; one in China in 1258)? No proxy- or documentary evidence available? Tree-ring and ice-core records are continuous and should have data in both 1258 and 1259. Please explain.

Inclusion of the available ice-core records from Antarctica would increase the database for the Southern Hemisphere (currently two records) substantially and better justify to speak of a "most complete globally resolved" database.

**L315-325:** Frost-rings in 1259 in SW USA support cold conditions during the growing season.

**L328:** Sulfate at GISP2 was analyzed at a biannual resolution. The age of the sample encompassing the rise and peak concentration value of 381 ppb is 1257.7 to 1259.6. Unless there is a higher resolved volcanic tracer available from GISP2 I can't see how one could constrain the peak sulfate fallout at this site to early 1259.

**Figure 4:** Maybe it would be helpful to also add the SAOD for Western Europe for the recommended PMIP4 forcing by Toohey & Sigl (2017) which is based on an assumed eruption date of July 1257.

**L347-358:** Your comparison focuses strongly on the timing of peak deposition of sulfate in both UKSEM1 scenarios versus that from ice cores. Based on previous aerosol-model inter-comparison projects it appears there is a large spread among state-of-the art climate models regarding the timing and hemispheric spread of sulfate across these climate models (Marshall et al., 2018) with this disagreement been linked to model physics and chemistry (Clyne et al., 2021). How sensitive are your specific results to the choice of the climate model? Would you get different results if you used a different climate model? Looking at Figure 9 in Marshall et al., (2018) it becomes evident that the spread of sulfate depositions all forced with the same $SO_2$ injection varies widely across the models, whereas sulfate deposition is rather comparable in timing, duration and magnitudes in ice cores from hemispheres.

[Figure]

**Figure 9.** Simulated area-mean volcanic sulfate deposition ($kg\,SO_4\,km^{-2}\,month^{-1}$) to the Antarctic ice sheet **(a)** and Greenland ice sheet **(b)** for each model (colours). Each ice sheet mean is defined by taking an area-weighted mean of the grid boxes in the appropriate regions once a land–sea mask has been applied. Solid lines mark the ensemble mean and shading is 1 SD. **(c)** Deposition fluxes from two monthly-resolved ice cores (DIV2010 from Antarctica and D4 from Greenland). The scale is reduced in **(c)**. The grey triangles mark the start of the eruption (1 April 1815).

**L352-355:** I don't fully understand this sentence, which has partly to do with the terminology. Resolution is the time interval contained in a sample (in this case monthly, assuming constant snowfall rate throughout the year); the dating precision is +/- 1 year, which means that one can shift the curve by adding or subtracting a year, but not 6 months as the dating is based on annual-layer counting of seasonal tracers. As noted above the mean age of the initial sulfate rise in four high-resolution ice cores (NGRIP, NEEM, WDC06A and Law Dome) is winter 1257/1258. Assuming a time lag of 6 months between the eruption in Indonesia and the start of sulfate rise in the polar ice cores (as was observed following Tambora 1815, Marshall et al., 2018) gives a best ice-core eruption age of summer 1257. A shift to summer 1256 or summer 1258 would be consistent with the ice-core age uncertainty, but inconsistent with the tree-ring cooling peaking in 1258.

**L355:** For better comparing the magnitude of sulfate deposition across hemispheres and across the two scenarios and any potential asymmetry in the deposition it would be helpful to provide a table with the total depositions from Samalas in models and ice cores.

It would also be helpful information to know to which extent the timing of initial sulfate, peak sulfate varies among different aerosol-climate models using for example the experiments done for Tambora.

**L380:** Maybe add: … with the bipolar ratio *of simulated sulfate depositions* between northern and …

**L384-387:** Here would be a good opportunity to put your results into context with a recent study (Guillet et al., 2023) using another aerosol-climate model (IPSL), tree rings and documentary evidence of lunar eclipses to constrain the age of the Samalas eruption (May to August 1257 is their most likely age).

**L407:** A stronger simulated cooling in the southern hemisphere (SH) must be expected since your model favors aerosol transport to the SH. With the absence of abundant well dated climate proxies for the SH (apart from ice cores in Antarctica) it will remain impossible to resolve the model/proxy disagreement previously discussed for the SH (Neukom et al., 2014).

**L410-411:** Since you invoke a potential connection between potential post-volcanic climate change and migration in the 13$^{th}$ century you may want to expand this to include two more large volcanic eruptions during this period in 1269 and 1275 both suspected to be located in the SH.

**L432:** Please consider removing this idea put forward on chronological errors in tree-ring chronologies or provide the right balance by also citing the numerous studies that have univocally rejected this speculation on globally missing tree rings (Anchukaitis et al., 2012; Buntgen et al., 2014; Esper et al., 2013; St. George et al., 2013).

**L438:** Stoffel et al. (2015) have reconstructed temperatures for the past 1,500 years and largely focused on the Samalas 1257 and Tambora 1815 eruptions. Other studies have extended the focus to the Common Era (Büntgen et al., 2020; Büntgen et al., 2016; Sigl et al., 2015)

**L459/462:** Annual resolution alone is not important; annual dating precision is the key requirement of a proxy in order to analyze volcanic climate impacts.

**L466-470:** A strong asymmetric distribution of sulfate is however not supported by the numerous ice-core records which show quite comparable magnitudes in deposition following Samalas and other known tropical eruptions (e.g. Tambora, Cosiguina, Krakatau) independent of their eruption season. (Table 1) How consistent is this interhemispheric asymmetry for different seasons across aerosol-climate models?

**Table1:** Ice-core indicated parameters of volcanic sulfate deposition for known tropical eruptions

| Event | Krakatau, 6°S | Cosigüina, 13°N | Tambora, 8°S | Samalas, 8°S | Source |
|---|---|---|---|---|---|
| Eruption Date | Aug 1883 | Jan 1835 | Apr 1815 | Jul(?) 1257 | |
| Sulfate Greenland [kg km$^{-2}$] | 18 | 19 | 38 | 105 | Toohey & Sigl 2017 |
| Sulfate Antarctica [kg km$^{-2}$] | 10 | 10 | 46 | 73 | Toohey & Sigl 2017 |
| Ratio Greenland/Antarctica | 1.7 | 2.0 | 0.8 | 1.4 | Toohey & Sigl 2017 |
| Start SO4 (NEEM) | 1883.6 | 1835.7 | 1815.6 | 1258.2 | Sigl et al., 2013 |
| Start SO4 (NGRIP) | 1883.5 | 1835.3 | 1816.1 | 1258.1 | Plummer et al., 2012 |
| Start SO4 (D4) | 1884.0 | 1835.0 | 1815.5 | N/A | (McConnell et al., 2007) ; Marshall et al., 2018 |
| Start SO4 (WDC06A) | 1884.0 | 1834.7 | 1815.4 | 1257.9 | Sigl et al., 2013 |
| Start SO4 (LawDome) | 1884.5 | 1836.7 | 1815.8 | 1257.4 | Plummer et al., 2012 |
| Start SO4 (DIV2010) | 1884.0 | 1835.8 | 1815.8 | N/A | (Sigl et al., 2014); Marshall et al., 2018 |
| Peak SO4 (NEEM) | 1884.3 | 1836.0 | 1816.3 | 1259.0 | Sigl et al., 2013 |
| Peak SO4 (NGRIP) | 1885.0 | 1836.2 | 1817.0 | 1259.4 | Plummer et al., 2012 |
| Peak SO4 (D4) | 1885.0 | 1836.2 | 1816.3 | N/A | McConnell et al., 2007; Marshall et al., 2018 |
| Peak SO4 (WDC06A) | 1885.0 | 1836.7 | 1816.5 | 1258.9 | Sigl et al., 2013 |
| Peak SO4 (LawDome) | 1885.0 | 1836.9 | 1817.0 | 1258.3 | (Jong et al., 2022) |
| Peak SO4 (DIV2010) | 1885.0 | 1836.9 | 1817.0 | N/A | Sigl et al., 2014; Marshall et al., 2018 |
| Dec. Date Eruption | 1883.6 | 1835.0 | 1815.3 | 1257.5 | |
| mean delta t (Greenland) [months] | 1.1 | 3.7 | 5.5 | 7.3 | Start of SO4 deposition minus eruption date |
| mean delta t (Antarctica) [months] | 6.7 | 8.3 | 4.5 | 1.3 | |

**L 475:** High temporal resolution is readily available for ice core records and volcanic eruptions located close-by or upwind of the ice sheets are already dated to the season using these ice cores (Veidivötn 1476/77, Paektu 946/47, Eldgja spring 939, Churchill 852/53, Okmok II 44/43 BCE; (Abbott et al., 2021; Mackay et al., 2022; McConnell et al., 2020; Oppenheimer et al., 2018)). The main limitation to estimate the timing of tropical eruptions stems from the limited number of observations from known tropical eruptions (see Table 1), variability in the transport time (1-8 months), and the disagreement in aerosol-climate models regarding stratospheric aerosol transport times between the tropics and the polar regions (see Marshall et al., 2018; Clyne et al. 2021).

**L491-496:** You may want to discuss here not only the potential but also the risks of using the agreement of model simulation output and proxies to constrain source parameters used as original forcing input; as one might easily run into a conformation bias, where one tends to favor solutions that best agree with our expectations.

The widely rejected "missing tree-ring hypothesis" (Mann et al., 2012) had its origin in a seemingly mismatch between climate simulations and proxy reconstructions which had subsequently been resolved (see PAGES 2k Consortium 2019, Nature Geoscience) by improved estimates of volcanic forcing (Toohey & Sigl 2007) and better climate reconstructions with inclusions of more tree-ring MXD records (Stoffel et al., 2015; Schneider et al. 2015; Wilson et al., 2016; Anchukaitis et al. 2017). In a similar way, the accuracy of ice-core dating has also been challenged because of an apparent mismatch between

strong tree-ring indicated cooling in the high-latitudes of the NH in 1453, and strong global volcanic forcing in 1458 (Esper et al., 2017). This hypothesis has also been rejected by recognizing evidence for two major eruptions 5 years apart and constraining the date of the larger signal in Greenland using tephra from an historic eruption in Iceland occurring in 1477 (Abbott et al., 2021). Both examples highlight the risk of using model-data agreement as the only diagnostic tracer for the correctness of the underlying records.

**L498-503:** I would omit or rephrase this section. The VSSI as reconstructed by ice cores is free of any climate response constraints, unlike for example previous reconstructions that had tuned the dates of volcanic forcing reconstructions (i.e. in 1453) to tree-ring indicated cooling extremes (Gao et al., 2008; Gao et al., 2006). Moreover, since the VSSI is in Toohey & Sigl (2017) based on 17 individual ice cores from both hemispheres a bias in the magnitude is much less an issue as for reconstructions that are based on single ice cores (Bader et al., 2020; Kobashi et al., 2017; Zielinski et al., 1994). Sampling stratospheric sulfate deposition in ice cores from both hemispheres simultaneously, is ideal for obtaining representative estimates of magnitude despite possible asymmetric sulfate distribution. To my knowledge, there are no known well dated large volcanic eruptions from the tropics which would not have been recognized as such in the polar ice-core records.

If in the future, existing disagreements in aerosol-climate models regarding the global spread of sulfate aerosols can be resolved it may eventually become possible to further constrain together with the existing high-resolution ice core records important source parameters such as the eruption season, latitude and/or plume heights. In any case it appears necessary to cite key studies analyzing the fate of stratospheric sulfur following major eruptions across the models (Clyne et al., 2021; Marshall et al., 2018; Quaglia et al., 2023).

**Additional References:**

Abbott, P. M., Plunkett, G., Corona, C., Chellman, N. J., McConnell, J. R., Pilcher, J. R., Stoffel, M., and Sigl, M.: Cryptotephra from the Icelandic Veiðivötn 1477 CE eruption in a Greenland ice core: confirming the dating of volcanic events in the 1450s and assessing the eruption's climatic impact, Clim. Past, 17, 565-585, 2021.

Anchukaitis, K. J., Breitenmoser, P., Briffa, K. R., Buchwal, A., Buntgen, U., Cook, E. R., D'Arrigo, R. D., Esper, J., Evans, M. N., Frank, D., Grudd, H., Gunnarson, B. E., Hughes, M. K., Kirdyanov, A. V., Korner, C., Krusic, P. J., Luckman, B., Melvin, T. M., Salzer, M. W., Shashkin, A. V., Timmreck, C., Vaganov, E. A., and Wilson, R. J. S.: Tree rings and volcanic cooling, Nat Geosci, 5, 836-837, 2012.

Bader, J., Jungclaus, J., Krivova, N., Lorenz, S., Maycock, A., Raddatz, T., Schmidt, H., Toohey, M., Wu, C.-J., and Claussen, M.: Global temperature modes shed light on the Holocene temperature conundrum, Nat Commun, 11, 4726, 2020.

Büntgen, U., Arseneault, D., Boucher, É., Churakova, O. V., Gennaretti, F., Crivellaro, A., Hughes, M. K., Kirdyanov, A. V., Klippel, L., Krusic, P. J., Linderholm, H. W., Ljungqvist, F. C., Ludescher, J., McCormick, M., Myglan, V. S., Nicolussi, K., Piermattei, A., Oppenheimer, C., Reinig, F., Sigl, M., Vaganov, E. A., and Esper, J.: Prominent role of volcanism in Common Era climate variability and human history, Dendrochronologia, 64, 125757, 2020.

Büntgen, U., Myglan, V. S., Ljungqvist, F. C., McCormick, M., Di Cosmo, N., Sigl, M., Jungclaus, J., Wagner, S., Krusic, P. J., Esper, J., Kaplan, J. O., de Vaan, M. A. C., Luterbacher, J., Wacker, L., Tegel, W., and Kirdyanov, A. V.: Cooling and societal change during the Late Antique Little Ice Age from 536 to around 660 AD, Nat Geosci, 9, 231-236, 2016.

Büntgen, U., Wacker, L., Nicolussi, K., Sigl, M., Gutler, D., Tegel, W., Krusic, P. J., and Esper, J.: Extraterrestrial confirmation of tree-ring dating, Nat Clim Change, 4, 404-405, 2014.

Clyne, M., Lamarque, J. F., Mills, M. J., Khodri, M., Ball, W., Bekki, S., Dhomse, S. S., Lebas, N., Mann, G., Marshall, L., Niemeier, U., Poulain, V., Robock, A., Rozanov, E., Schmidt, A., Stenke, A., Sukhodolov, T., Timmreck, C., Toohey, M., Tummon, F., Zanchettin, D., Zhu, Y., and Toon, O. B.: Model physics and chemistry causing intermodel disagreement within the VolMIP-Tambora Interactive Stratospheric Aerosol ensemble, Atmos. Chem. Phys., 21, 3317-3343, 2021.

Esper, J., Buntgen, U., Hartl-Meier, C., Oppenheimer, C., and Schneider, L.: Northern Hemisphere temperature anomalies during the 1450s period of ambiguous volcanic forcing, B Volcanol, 79, 2017.

Esper, J., Buntgen, U., Luterbacher, J., and Krusic, P. J.: Testing the hypothesis of post-volcanic missing rings in temperature sensitive dendrochronological data, Dendrochronologia, 31, 216-222, 2013.

Gao, C. C., Robock, A., and Ammann, C.: Volcanic forcing of climate over the past 1500 years: An improved ice core-based index for climate models, J Geophys Res-Atmos, 113, 2008.

Gao, C. C., Robock, A., Self, S., Witter, J. B., Steffenson, J. P., Clausen, H. B., Siggaard-Andersen, M. L., Johnsen, S., Mayewski, P. A., and Ammann, C.: The 1452 or 1453 AD Kuwae eruption signal derived from multiple ice core records: Greatest volcanic sulfate event of the past 700 years, J Geophys Res-Atmos, 111, 2006.

Guillet, S., Corona, C., Oppenheimer, C., Lavigne, F., Khodri, M., Ludlow, F., Sigl, M., Toohey, M., Atkins, P. S., Yang, Z., Muranaka, T., Horikawa, N., and Stoffel, M.: Lunar eclipses illuminate timing and climate impact of medieval volcanism, Nature, 616, 90-95, 2023.

Jong, L. M., Plummer, C. T., Roberts, J. L., Moy, A. D., Curran, M. A. J., Vance, T. R., Pedro, J. B., Long, C. A., Nation, M., Mayewski, P. A., and van Ommen, T. D.: 2000 years of annual ice core data from Law Dome, East Antarctica, Earth Syst. Sci. Data, 14, 3313-3328, 2022.

Kobashi, T., Menviel, L., Jeltsch-Thommes, A., Vinther, B. M., Box, J. E., Muscheler, R., Nakaegawa, T., Pfister, P. L., Doring, M., Leuenberger, M., Wanner, H., and Ohmura, A.: Volcanic influence on centennial to millennial Holocene Greenland temperature change, Sci Rep-Uk, 7, 2017.

Mackay, H., Plunkett, G., Jensen, B. J. L., Aubry, T. J., Corona, C., Kim, W. M., Toohey, M., Sigl, M., Stoffel, M., Anchukaitis, K. J., Raible, C., Bolton, M. S. M., Manning, J. G., Newfield, T. P., Di Cosmo, N., Ludlow, F., Kostick, C., Yang, Z., Coyle McClung, L., Amesbury, M., Monteath, A., Hughes, P. D. M., Langdon, P. G., Charman, D., Booth, R., Davies, K. L., Blundell, A., and Swindles, G. T.: The 852/3 CE Mount Churchill eruption: examining the potential climatic and societal impacts and the timing of the Medieval Climate Anomaly in the North Atlantic region, Clim. Past, 18, 1475-1508, 2022.

Marshall, L., Schmidt, A., Toohey, M., Carslaw, K. S., Mann, G. W., Sigl, M., Khodri, M., Timmreck, C., Zanchettin, D., Ball, W. T., Bekki, S., Brooke, J. S. A., Dhomse, S., Johnson, C., Lamarque, J. F., LeGrande, A. N., Mills, M. J., Niemeier, U., Pope, J. O., Poulain, V., Robock, A., Rozanov, E., Stenke, A., Sukhodolov,

T., Tilmes, S., Tsigaridis, K., and Tummon, F.: Multi-model comparison of the volcanic sulfate deposition from the 1815 eruption of Mt. Tambora, Atmos. Chem. Phys., 18, 2307-2328, 2018.

McConnell, J. R., Edwards, R., Kok, G. L., Flanner, M. G., Zender, C. S., Saltzman, E. S., Banta, J. R., Pasteris, D. R., Carter, M. M., and Kahl, J. D. W.: 20th-century industrial black carbon emissions altered arctic climate forcing, Science, 317, 1381-1384, 2007.

McConnell, J. R., Sigl, M., Plunkett, G., Burke, A., Kim, W. M., Raible, C. C., Wilson, A. I., Manning, J. G., Ludlow, F., Chellman, N. J., Innes, H. M., Yang, Z., Larsen, J. F., Schaefer, J. R., Kipfstuhl, S., Mojtabavi, S., Wilhelms, F., Opel, T., Meyer, H., and Steffensen, J. P.: Extreme climate after massive eruption of Alaska's Okmok volcano in 43 BCE and effects on the late Roman Republic and Ptolemaic Kingdom, P Natl Acad Sci USA, 117, 15443-15449, 2020.

Neukom, R., Gergis, J., Karoly, D. J., Wanner, H., Curran, M., Elbert, J., Gonzalez-Rouco, F., Linsley, B. K., Moy, A. D., Mundo, I., Raible, C. C., Steig, E. J., van Ommen, T., Vance, T., Villalba, R., Zinke, J., and Frank, D.: Inter-hemispheric temperature variability over the past millennium, Nat Clim Change, 4, 362-367, 2014.

Oppenheimer, C., Orchard, A., Stoffel, M., Newfield, T. P., Guillet, S., Corona, C., Sigl, M., Di Cosmo, N., and Büntgen, U.: The Eldgja eruption: timing, long-range impacts and influence on the Christianisation of Iceland, Climatic Change, 147, 369-381, 2018.

PAGES2k Consortium: Data Descriptor: A global multiproxy database for temperature reconstructions of the Common Era, Sci Data, 4, 2017.

Plummer, C. T., Curran, M. A. J., van Ommen, T. D., Rasmussen, S. O., Moy, A. D., Vance, T. R., Clausen, H. B., Vinther, B. M., and Mayewski, P. A.: An independently dated 2000-yr volcanic record from Law Dome, East Antarctica, including a new perspective on the dating of the 1450s CE eruption of Kuwae, Vanuatu, Clim Past, 8, 1929-1940, 2012.

Quaglia, I., Timmreck, C., Niemeier, U., Visioni, D., Pitari, G., Brodowsky, C., Brühl, C., Dhomse, S. S., Franke, H., Laakso, A., Mann, G. W., Rozanov, E., and Sukhodolov, T.: Interactive stratospheric aerosol models' response to different amounts and altitudes of SO2 injection during the 1991 Pinatubo eruption, Atmos. Chem. Phys., 23, 921-948, 2023.

Salzer, M. W. and Hughes, M. K.: Bristlecone pine tree rings and volcanic eruptions over the last 5000 yr, Quaternary Res, 67, 57-68, 2007.

Sigl, M., McConnell, J. R., Layman, L., Maselli, O., McGwire, K., Pasteris, D., Dahl-Jensen, D., Steffensen, J. P., Vinther, B., Edwards, R., Mulvaney, R., and Kipfstuhl, S.: A new bipolar ice core record of volcanism from WAIS Divide and NEEM and implications for climate forcing of the last 2000 years, J Geophys Res-Atmos, 118, 1151-1169, 2013.

Sigl, M., McConnell, J. R., Toohey, M., Curran, M., Das, S. B., Edwards, R., Isaksson, E., Kawamura, K., Kipfstuhl, S., Kruger, K., Layman, L., Maselli, O. J., Motizuki, Y., Motoyama, H., Pasteris, D. R., and Severi, M.: Insights from Antarctica on volcanic forcing during the Common Era, Nat Clim Change, 4, 693-697, 2014.

Sigl, M., Winstrup, M., McConnell, J. R., Welten, K. C., Plunkett, G., Ludlow, F., Büntgen, U., Caffee, M., Chellman, N., Dahl-Jensen, D., Fischer, H., Kipfstuhl, S., Kostick, C., Maselli, O. J., Mekhaldi, F., Mulvaney,

R., Muscheler, R., Pasteris, D. R., Pilcher, J. R., Salzer, M., Schupbach, S., Steffensen, J. P., Vinther, B. M., and Woodruff, T. E.: Timing and climate forcing of volcanic eruptions for the past 2,500 years, Nature, 523, 543-549, 2015.

St. George, S., Ault, T. R., and Torbenson, M. C. A.: The rarity of absent growth rings in Northern Hemisphere forests outside the American Southwest, Geophys Res Lett, 40, 3727-3731, 2013.

Zielinski, G. A., Mayewski, P. A., Meeker, L. D., Whitlow, S., Twickler, M. S., Morrison, M., Meese, D. A., Gow, A. J., and Alley, R. B.: Record of Volcanism since 7000-Bc from the Gisp2 Greenland Ice Core and Implications for the Volcano-Climate System, Science, 264, 948-952, 1994.

---

## Author Comment (AC2)

**Response to Editor and reviewer comments on "Utilizing a Multi-Proxy to Model Comparison to**
**Constrain the Season and Regionally Heterogeneous Impacts of the Mt. Samalas 1257 Eruption" by**
**Wainman et al .**

**Editor Comments**

You will see that you have two expert reviews on your paper. Both like the ideas in your paper and
consider it well-written, but both propose major revisions. You should respond to all the comments
in both reviews. Once you have done this I will be asked to give a final decision as editor but I
anticipate asking you to prepare a new version based on the comments. Please do be sure to
address the most substantive comments in the reviews. I am particularly concerned that both
reviewers wonder about using a single model, and their comments question whether it is reasonable
to constrain the date of the eruption based on the results of a single model and only two time
points. This seems like quite a strong concern, and you may wish to tone down the certainty of some
of your conclusions if you cannot address it with additional runs or information from additional
models.

We would like to warmly thank both reviewers and the editor for their very constructive comments
which have greatly improved the quality of the manuscript, as well as their patience in our time to
fully respond to their comments. We have addressed the comments in full, and below we provide a
detailed line-by-line response to each comment. We also provide a new version of the manuscript that
shows all related changes.

During our preprint being open for comments online we also received additional correspondence from
Prof Monica Green who helpfully added an additional perspective on the historical implications and
plausible connections for the Samalas eruption. We have therefore updated L69-73 to reflect this:

*"Suggestions that the Samalas eruption can be linked to the initiation of the "Big Bang" diversification*
*event which led to the Branch 1 strain of Yersinia pestis responsible for the Black Death in Europe (Fell*
*et al., 2020) have recently been refuted, with consensus forming instead that this plague proliferation*
*event can be traced to the Tian Shan region much earlier in the 13th century (Green, 2020, Green,*
*2022). Nonetheless, connections have still been drawn between the anomalous climatic conditions*
*following the eruption and the fall of Bagdad to the Mongol empire in 1258, as well as the subsequent*
*defeat of the Mongol Army at the battle of Ayn Jālūt in 1260 which marked the collapse of the Mongol*
*westward advance (Green, 2020, Di Cosmo et al., 2021). Without a comprehensive understanding of*
*the extent and chronology of climate response to the eruption on a regional scale, the robustness of*
*these inferred connections between post-eruption climate response and historical events remains*
*difficult to constrain."*

**Reviewer 1**

Review: "Utilizing a Multi-Proxy to Model Comparison to Constrain the Season and Regionally
Heterogeneous Impacts of the Mt. Samalas 1257 Eruption" by Laura Wainman et al.

We thank reviewer 1 for their constructive and insightful comments, please find our responses to your
suggestions below.

**A: General comments:**

The authors aim to further constrain the date of the mid-13th century Samalas eruption in
Indonesia, the largest volcanic eruption in the Common Era. For this they use ensembles of aerosol-
climate model simulations forced with SO2 injections in either July 1257 or January 1258, the most widely used dates for this eruption. They find that the simulated global climate response for the July
1257 eruption scenario produces better agreement with collated proxy reconstructions for 1257 to
1259 and argue on those grounds that the eruption likely occurred in summer 1257, in agreement
with previous papers.

Tropical eruptions often remain difficult to date exactly, because of a lack of documentary records
from direct observations and the scarce available information preserved under tropical climates in
the areas surrounding the volcanos. The seasonal timing of eruptions can have important effects on
the global climate impact. This paper is a welcome contribution to ongoing efforts in this research
field. It is well written, with a clear structure and outline of the scientific hypothesis, especially
considering this is a master thesis. The setup of the aerosol-climate model experiments appears
appropriate (for a non-expert) and I acknowledge that the authors also have collated and included a
large set of proxy records from different archives. I would like to invite the authors to address in
their revision a number of points (outlined in detail under Specific Comments) focusing on three
main topics:

1)    The framing: The previous arguments and references used to argue for a 1258 eruption
are somewhat outdated going back to the early 2000s when ice-core records were
biannually resolved, large-scale tree-ring records were not fully developed and the
source of the eruption had been unknown; in the last years the research community has
already largely converged towards accepting a date in 1257, which wasn't really
challenged.

Since the beginning of this work (3 years ago) consensus has more clearly converged on a Summer
1257 date for the Mt Samalas eruption and as such we have altered the framing in the introduction
and abstract to reflect that more clearly (see lines 12-15 and 46-63 in the revised manuscript).
However, whilst consensus has converged there are still limitations in the evidence proposed to
support a 1257 date: 1) radiocarbon dating of PDC material only provides an earliest eruption
boundary of 1257 but does not exclude a 1258 eruption date (Lavigne et al., 2013), 2) whilst there is
documentary evidence for mild January 1258 conditions, the link between large tropical eruptions and
NH winter warming remains disputed (Polvani and Camargo, 2020) 3) Only 26 out of 170 tree ring
chronologies presented by Buntgen et al., 2022 show negative TRW anomalies in North America for
1257 with widespread summer cooling only becoming pronounced in 1258. Therefore, we suggest
that our work, using UKESM and a combination of multi-proxy evidence, is able to distinguish more
robustly between Summer 1257 and January 1258 eruption dates. Our work also serves as a case study
for applying the model-multi proxy approach to constraining eruption source parameters, and serves
to highlight both the usefulness and potential pitfalls of this approach.

2)    The rationale: A more general discussion would be welcome to which extent we wanted
(or should avoid) to constrain external climate forcing through comparisons of simulated
and reconstructed climate response. I noted two specific examples below in which the
desire for a high model-proxy agreement has led to speculations of dating errors in tree-
rings and in ice cores. In both cases these speculations were later rejected based on
climate-independent geochronological data (i.e. radiocarbon, tephra). So there is an
eminent risk of overfitting climate forcing records by aiming to minimize model-data
disagreement. A short discussion on the risks (beside the potential) would be helpful.

We agree that this is an important and ongoing discussion and have added a paragraph to the
discussion section to reflect this. We thank the reviewer for highlighting these specific case studies.

*Lines 568 to 583 in the revised version of the manuscript: "Whilst comparisons of simulated and reconstructed climate responses following large volcanic eruptions have been used routinely and effectively by a multitude of studies (van Dijk et al., 2023, Büntgen et al., 2022, Stoffel et al., 2015 ), there are several limitations to this approach. Given the uncertainties associated with both proxy reconstructions and model simulations neither can be taken as the inherently "correct" baseline with which to fit the other, and thus particular care should be taken when using model-proxy comparison to validate the correctness of underlying records or model input parameters. For example, the missing tree ring hypothesis (Mann et al., 2012), which has since been widely rejected, proposed that the mismatch between climate simulations and proxy reconstructions resulted from chronological errors due to missing growth rings. This has, however, since been resolved with improved estimates of volcanic forcing (Toohey & Sigl 2017), the inclusion of climate-independent geochronological data, and the greater inclusion of MXD records in tree ring reconstructions (Stoffel et al., 2015; Schneider et al. 2015; Wilson et al., 2016; Anchukaitis et al. 2017). When utilising an array of different proxy data there is the risk of confirmation bias meaning records which show significant agreement with model simulations being given greater weighting than those which show less agreement. A more quantifiable approach to model-proxy comparison, where a greater number of model realisations would allow for more robust statistical evaluation would therefore be a considerable improvement for the application of the model-multi proxy framework. Further to this, better uncertainty quantification for proxy data would enable more robust comparison with model outputs, particularly in multi-model comparison studies."*

The implications. I have some reservations regarding your interpretation of the ice-core records (outlined in detail below) and also the implications this study has for the use of ice cores to infer volcanic forcing in the past. Existing limitations to constrain the exact timing of tropical eruptions are dominated by the poorly constrained time-lag of sulfate deposition on the ice sheets following a volcanic eruption, and to a less extent by the resolution of the ice-core records. This time-lag is also highly variable among different state-of-the-art aerosol models. I therefore argue that the resulting spread in stratospheric aerosols and the spatio-temporal climate fingerprint will depend foremost on the choice of the aerosol-climate model. Given these model uncertainties I currently do not see a strong case to constrain the climate-independent ice-core estimates of volcanic forcing with one specific aerosol-climate model.

We acknowledge the limitations of our initial conclusions based on the comparison of ice core records with our model simulated sulfate deposition. Upon reflection we have removed this section (Lines 471 - 476 in the original version of the manuscript) and instead focussed our discussion on the model-specific dependencies of our results and current limitations in comparing ice core and model simulated results given the large inter-model variations which have been demonstrated by previous studies (Marshall et al., 2018, Quaglia et al., 2023) - see lines 549-569 in the revised version of the manuscript.

**B: Specific comments:**

L35: I doubt that these two references provided the necessary record length, spatial representation and completeness to put the size of the sulfate deposition into a 2500-year context.

A reference to Sigl et al., 2015 was missing here and has been added. This study spans 2,500 years with an array of Greenland and Antarctic ice cores, where 1258 has the largest non-sea-salt Sulfur (nssS) deposition of the 2,500 year study period.

L40: please replace Wade et al. (2020) with Schneider et al., (2015).

This has been corrected.

L45: I am not aware of anybody citing January 1258 as a potential date, and the only citation in this
paragraph (Stothers 2000) speaks of early 1258. Please clarify.

Please see our response to the question of framing above (Lines 63-76 in this document).

We have updated lines 46-49 in the revised version of the manuscript: "*The full span of dates proposed*
*for the Mt Samalas eruption ranges from 1256 to 1258, with suggestions including an eruption in*
*spring 1256 (Bauch, 2019), summer 1257 (Lavigne et al., 2013, Oppenheimer 2003), and early 1258*
*(Stothers 2000). Whilst consensus has converged on a summer 1257 eruption date, as of yet, no single*
*combination of evidence has been able to robustly distinguish between, and exclude other dates*
*proposed for the Mt Samalas eruption.*"

L51-52: I find it unlikely that any radiocarbon date on pyroclastic material would provide the
necessary precision to contribute to the discussion of the actual calendar year of the eruption. All
the more so as the paper describing it is already 20 years old.

A reference to Lavigne et al., 2013 has been added here. They use radiocarbon and calibrated ages of
the charcoal samples from the Samalas pyroclastic density current deposits (using OxCal 4.2.2 and
IntCal 09). Although some samples are older, no samples show ages younger than 1257 and thus they
define a younger eruption age boundary of A.D 1257. We have revised the manuscript as follows:

Lines 52-55: "*A mid-1257 eruption date was first proposed by Oppenheimer (2003) based on the spatial*
*distribution of negative temperature anomalies across both hemispheres for 1257-59. Radiocarbon*
*dating of the pyroclastic flow deposits associated with the eruption also yield a youngest eruption age*
*boundary of 1257, with some samples suggesting an earlier eruption date, but no samples suggesting*
*a date later than 1257 (Lavigne et al., 2013).*"

L54: Besides TRW anomalies the Western US tree-rings (Salzer and Hughes, 2007) also had frost rings
in 1257 and 1259 which combined provide strong evidence of a volcanic source for the cooling and
which may help to constrain the age of the eruption.

Reference to these frost rings has been added to line 58:

*"Negative tree-ring width (TRW) growth anomalies in the late 1257-growth season  (Büntgen et al.,*
*2022) and frost rings (Salzer and Hughes, 2007) in the Western US in 1257 and 1259  also add*
*support for a potential eruption date prior to August 1257."*

The absence of frost rings in 1258 is also of  interest, given this contradicts model simulated cooling
which is strongest in 1258 across our model simulated SAT anomalies. This has been added to the
discussion section on line 461:

"*The presence of frost rings  in 1257 and 1259 (Salzer and Hughes, 2007), but not in 1258, also*
*contradicts model-simulated cooling, which consistently shows the strongest cooling in 1258.*"

L57: The statement "peak sulfate deposition in ice cores is recorded for 1259" lacks a reference and
it is also not true for all ice cores. Some ice cores have the peak sulfate levels in January 1259
(NEEM), others in November 2018 (WDC06A). See Table 1 below. For the timing of the eruption the
timing of the peak sulfate level is anyhow not of great relevance but rather the timing of the start of
volcanic sulfate deposition which is dated between 1257.4 and 1258.2 in four different records from
Greenland and Antarctica (Plummer et al., 2012; Sigl et al., 2013).

The intention of this sentence was to explain what evidence led to the original proposal by Stothers
(2000) for an early 1258 eruption date. We have amended the sentence to make this clearer:

Line 61: "*Nonetheless, Stothers (2000) suggests a later eruption date of early 1258 based on peak*
*sulfate deposition for the Mt. Samalas eruption occurring in 1259 (from Hammer et al., 1980) and the*
*first historical reports of a dust veil over Europe appearing in Summer 1258, which they suggest is*
*most compatible with an early 1258 eruption.*"

We also thank the reviewer for collating Table 1. This has greatly aided our later discussion of
simulated sulfate deposition and the comparison of our two eruption scenarios with ice core
deposition records. See our responses below.

L90: How would spring (April) or fall (September) eruption scenario differ from January and July?

Whilst additional model simulations for a Spring or Autumn eruption date would certainly be
valuable, it is not possible to run any further simulations as part of this project. Summer and winter
realisations were chosen as end-member scenarios to emphasise the effects of annual variation in
atmospheric circulation. The suggested dates for the Samalas eruption specifically also fall into
either Summer or Winter eruption scenarios, and so given model resources these were the priority
to test.

L92: I would refrain from calling the database most complete; for your research question you don't
need the most complete dataset but the most suited data.

We have removed this sentence.

L140: Your selection criteria should not be annual resolution but annual age precision. Only tree-
rings and documentary records fulfil this requirement and all ice cores in which the Samalas eruption
had been securely identified. For these reasons, I would strongly advise you to remove the lake
sediment records and the Svalbard record from your analysis all of which contain no clear signatures
of Samalas and are not dated to 1258 (±0). On the other hand, many more than the three used ice
cores (GRIP, CRETE, DYE3) would be readily available for analysis both from Greenland and from
Antarctica.

Lake Sediment and Svalbard records have been removed on the basis of insufficient age dating
precision. 6 additional ice core records (2 from Greenland, 4 from Antarctica) have been
incorporated. These records have both annual resolution and an age dating precision of +/- 1 yr. See
Supplementary Sheet 1 under "Ice Cores" for record metadata. Linear regression analysis was
applied to calibrate the ice core δ18O series to JJA gridded temperature anomalies (with respect to
1990-1960) from the BEST dataset (Rodhe et al., 2020).

L157-159: A more comprehensive dataset from Greenland would potentially be more skillful than a
set of only three ice cores. The PAGES2k Consortium (2017) database contains at least 9 ice cores
from both Greenland and Antarctica with annual resolution encompassing the age of Samalas
(PAGES2k Consortium 2017).

Additional ice core records from Greenland and Antarctica have been incorporated (see response
above). The ice core methods section has also been updated to reflect this:

Lines 165-173: "We include six δ18O isotope series ice core records from Greenland and Antarctica,
where records were chosen on the basis of both annual resolution and an age dating precision of +/-

1 year. Linear regression analysis was applied to calibrate the series to JJA gridded temperature
anomalies (with respect to 1990-1960) from the BEST dataset (Rodhe et al., 2020). An additional SAT
constraint is also included in Greenland for Summer 1258 from analysis by Guillet et al., (2017) who
utilised three Greenland ice cores at GRIP, CRETE, and DYE3 (Vinther et al., 2010) to calculate a
clustered SAT anomaly for the region. Additional ice core records were investigated to expand this
analysis such as the Illimani Ice Core in Bolivia and the Belukha Ice Core in Altai, Siberia; however,
these records lacked the annual resolution required to constrain abrupt temperature changes
associated with volcanic eruptions and/or the age dating precision to clearly identify signatures from
the Samalas eruption. ”

L159-169: As stated above I don't think that any of these climate information provided for these
records can with certainty be linked to the years 1257-59.

See response above to comment on L140.

L276: Please cite the original reference (Vinther et al., 2010).

This reference has been added.

Figure 3: What does the absence of symbol indicate (e.g. Japan and Greenland in 1259; one in China
in 1258)? No proxy- or documentary evidence available? Tree-ring and ice-core records are
continuous and should have data in both 1258 and 1259. Please explain.

Figure 3 has been remade to incorporate the six additional ice core records. The figure has also been
split into two - Figure 3 now in the main text includes only the model-proxy comparison panels, whilst
the proxy distribution map has been moved to the supplementary (Figure S3). In Japan the Square
symbol present in 1258 refers to the historical record (Azuma Kagami - Farris, 2006) which only makes
reference to abnormal weather conditions in 1258, therefore the symbol is absent for 1259 as the
documentary evidence is not available. The record in China (Altai Mountains) is also a historical source
which makes reference to abnormal snowfall in the region in July 1259. Guillet et al., 2017 only
provides a value for their Greenland ice core cluster for 1258.

[Figure]

*Figure 3: Globally-resolved multi-proxy-model comparison visualized for summers (JJA) 1258 and 1259.*
*Symbols denote proxy data type and red/blue shading shows model-simulated surface air temperature*
*anomalies for JUL1257 eruptions (a-b) and JAN1258 eruptions (c-d) ensemble means. Surface air temperature*
*anomalies were calculated relative to a 10-year background climatology constructed from the control ensemble*
*mean. Hashed lines denote anomalies at <95% significance as determined by a grid point ANOVA analysis.*
*Black filled symbols denote agreement within +/- 1°C between model-simulated anomalies and quantitative*
*proxy records. Grey filled symbols denote qualitative agreement with proxy records. Locations and proxy-*
*constrained SAT anomalies are shown in Figure S3.*

Inclusion of the available ice-core records from Antarctica would increase the database for the
Southern Hemisphere (currently two records) substantially and better justify to speak of a "most
complete globally resolved" database.

additional Ice Core records from Antarctica have been incorporated and thus significantly expand
the proxy network across the Southern Hemisphere.

L315-325: Frost-rings in 1259 in SW USA support cold conditions during the growing season.

These frost rings have been added to Figure 3. It is of interest to highlight the presence of frost rings
in 1259 but their absence in 1258 - this has been added subsequently to our discussion (see also
response to comment on L54 above).

L328: Sulfate at GISP2 was analyzed at a biannual resolution. The age of the sample encompassing
the rise and peak concentration value of 381 ppb is 1257.7 to 1259.6. Unless there is a higher
resolved volcanic tracer available from GISP2 I can't see how one could constrain the peak sulfate
fallout at this site to early 1259.

The reference here has been corrected to Hammer et al., 1980 which uses the Crete ice core which is
dated to +/- 1 year accuracy for the past 900 years and is the record Stothers 2000 refers to.

Figure 4: Maybe it would be helpful to also add the SAOD for Western Europe for the recommended
PMIP4 forcing by Toohey & Sigl (2017) which is based on an assumed eruption date of July 1257.

SAOD for PMIP4 forcing (Toohey and Sigl, 2017) has been added to Figure 4.

L347-358: Your comparison focuses strongly on the timing of peak deposition of sulfate in both
UKSEM1 scenarios versus that from ice cores. Based on previous aerosol-model inter-comparison
projects it appears there is a large spread among state-of-the art climate models regarding the
timing and hemispheric spread of sulfate across these climate models (Marshall et al., 2018) with
this disagreement been linked to model physics and chemistry (Clyne et al., 2021). How sensitive are
your specific results to the choice of the climate model? Would you get different results if you used a
different climate model? Looking at Figure 9 in Marshall et al., (2018) it becomes evident that the
spread of sulfate depositions all forced with the same $SO_2$ injection varies widely across the models,
whereas sulfate deposition is rather comparable in timing, duration and magnitudes in ice cores
from hemispheres.

This is a very good point that was not discussed sufficiently in the previous version of the manuscript.
Marshall et al., 2018 show considerable variability in the timing, magnitude, and spatial distribution
of sulfate deposition between climate models for the Tambora 1815 eruption (see Table S3) . A fuller
discussion of this variability and associated uncertainty has been added to our discussion on model limitations (lines 549-568 in the revised manuscript). Nonetheless, despite variability between models,
this is unlikely to affect the considerable offset we see in the timing of both onset and peak sulfate
deposition between July and January eruption scenarios for our model simulations, on which our
comparison to ice core records in based (where agreement between ice core records is far more
consistent). However, the magnitude and hemispheric asymmetry of sulfate deposition does appear
to be strongly model dependent. For this reason we do not base our model-ice core comparison on
the absolute magnitude or asymmetry of simulated sulfate deposition. Greenland/Antarctica ratios
are discussed, however, the model-dependency of sulfate distribution and associated limitations is
also highlighted in the discussion as follows:

Lines 549-568 "*Nonetheless, model simulations of the Tambora 1815 eruption by Marshall et al., 2018*
*found a strong model dependency for the timing, magnitude, and spatial distribution of sulfate*
*deposition. Polar sulfate deposition in a previous version of the UK climate model (UM-UKCA) was half*
*that reconstructed using ice cores for the Tambora 1815 eruption and considerably lower than*
*deposition in the other MAECHAM5-HAM and SOCOL-AER models analysed (See Table S3). In our*
*JUL1257 and JAN1258 simulations for the Samalas 1257 eruption mean total deposition in Antarctica*
*(55-28 Kg km-2) and Greenland (26-23 Kg km-2) was also lower than mean total deposition from ice*
*cores records (73 and 105 Kg km-2  respectively, see Table 2). The overall lower polar total sulfate*
*deposition in the UKESM may be caused by too weak polarward transport (or stronger meridonal*
*deposition) (Marshall et al., 2018). For our JUL1257 and JAN1258 model-simulated sulfate deposition*
*for both eruption scenarios result in Greenland to Antarctica ratios < 1, with a Summer 1257 eruption*
*showing a much more asymmetric distribution (0.49) than a January 1258 eruption (0.83). These ratios*
*however, disagree with the Greenland to Antarctica ratio derived from ice cores (1.4, Toohey and Sigl,*
*2017) which if the ice core values are robust, suggests that for our Samalas scenarios, sulfate aerosol*
*transport to the NH is too low, thus favouring transport to the SH. Of the four models analysed in*
*Marshall et al., (2018) for the Tambora 1815 eruption, UM-UKCA was one of two models that had*
*greater deposition in Greenland compared to Antarctica (Table S3, Greenland/Antarctica = 1.7). Both*
*the Tambora 1815 and Mt Samalas 1257 were large magnitude eruptions at a similar latitude,*
*therefore this intra-model difference in the asymmetric distribution of sulfate aerosol most likely*
*results from differences in initial conditions used for our simulations (such as the phase of the QBO).*
*Overall, this further highlights the complications of disentangling inter-model differences and intra-*
*model variation due to initial conditions. Given both these inter and intra-model differences in relative*
*hemispheric aerosol distribution and deposition, the current robustness of using hemispheric sulfate*
*deposition ratios to distinguish between eruption scenarios when compared to ice core records is*
*limited, with further work being needed to understand model and starting condition specific effects.* "

L352-355: I don't fully understand this sentence, which has partly to do with the terminology.
Resolution is the time interval contained in a sample (in this case monthly, assuming constant
snowfall rate throughout the year); the dating precision is +/- 1 year, which means that one can shift
the curve by adding or subtracting a year, but not 6 months as the dating is based on annual-layer
counting of seasonal tracers. As noted above the mean age of the initial sulfate rise in four high-
resolution ice cores (NGRIP, NEEM, WDC06A and Law Dome) is winter 1257/1258. Assuming a time
lag of 6 months between the eruption in Indonesia and the start of sulfate rise in the polar ice cores
(as was observed following Tambora 1815, Marshall et al., 2018) gives a best ice-core eruption age
of summer 1257. A shift to summer 1256 or summer 1258 would be consistent with the ice-core age
uncertainty, but inconsistent with the tree-ring cooling peaking in 1258.

[Figure]

To address this comment, we have revised the manuscript as follows:

Lines 364 - 390: "Figures 4B and 4C show ice sheet-averaged model-simulated sulfate deposition
across Greenland and Antarctic ice sheets respectively, for JUL1257 and JAN1258 eruption scenarios.
The mean timing of the onset of sulfate rise and the timing of peak sulfate deposition across four high
resolution ice cores (see Table S2) are also shown as vertical black lines. Table 2 shows a comparison
between ice core average (from Table S2) and ensemble means for JUL1257 and JAN1258 model
simulations for the timing of SO4 rise, peak, and total deposition in Greenland and Antarctica. The
JUL1257 ensemble shows that the onset  of sulfate deposition occurs in January 1285 in Greenland
and October 1257 in Antarctica, with peak deposition occurring in July 1258 and February 1258
respectively.  By contrast, the JAN1258 ensemble shows sulfate rise beginning in May and March 1258
in Greenland and Antarctica respectively, with peak deposition in Jan 1259 and September 1258.
Across four high resolution ice cores (Table S2) the mean timing of the onset of sulfate deposition
occurs in February 1258 in Greenland and August 1257 in Antarctica, thus showing closest agreement
with the timing of the simulated onset in sulfate deposition for a July 1257 eruption date, where only
an eruption in summer 1257 can account for the beginning of sulfate deposition in Antarctica in
autumn 1257. Across the four ice core records mean peak deposition in Greenland occurs in March
1259 whilst mean peak deposition in Antarctica occurs earlier in July 1258. The timing of peak
deposition therefore shows better agreement with simulated peak deposition for a January 1258
eruption date with peak model-simulated deposition occurring too early in the JUL1257 ensemble
relative to ice core records. Ice core records also suggest an asymmetry in sulfate dispersal and
deposition, with the onset of sulfate deposition and peak deposition in Antarctica being 6 and 8
months ahead of Greenalnd respectively. The JUL1257 ensemble shows a greater degree of asymmetry compared to the JAN1258 ensemble, although offsets of 3 and 5 months between
Antarctica and Greenland are still lower than the offset suggested by the ice core record. Whilst the
offset between the onset in sulfate deposition and peak sulfate deposition is approximately a year in
both Greenland (13 months) and Antarctica (11 months), the offset for model-simulated deposition in
the UKESM is considerably shorter across both JUL1257 and JAN1258 ensembles (ranging from 4-8
months). This may represent a limitation specific to the UKESM, for example with too weak poleward
transport of volcanic aerosol (or midlatitude deposition too strong). This weaker poleward transport
may also contribute to the lower magnitude of total polar deposition simulated by the UKESM relative
to ice core records, also seen in an earlier version of the UK climate model for the 1815 eruption of
Mt. Tambora (Marshall et al., 2018). Both model-simulated ensembles show greater total sulfate
deposition in Antarctica compared to Greenland, whilst ice core records suggest the opposite
asymmetry favouring greater deposition in Greenland. "

L355: For better comparing the magnitude of sulfate deposition across hemispheres and across the
two scenarios and any potential asymmetry in the deposition it would be helpful to provide a table
with the total depositions from Samalas in models and ice cores. It would also be helpful information
to know to which extent the timing of initial sulfate, peak sulfate varies among different aerosol-
climate models using for example the experiments done for Tambora.

Three additional tables have been added (and are referred to in our response to L352-355 above):

-   Table S2: Shows the timing of $SO_4$ rise and peak in 4 ice cores from Greenland and Antarctica
as well as total sulfate deposition and Greenland/Antarctica ratio from ice core records.
-   Table 2: Shows the mean timing of $SO_4$ rise and peak in JUL1257 and JAN1258 model
simulations compared to ice core means (from Table S2) as well as total sulfate deposition
and Greenland/Antarctica ratios.
-   Table S3: Reproduced from Marshall et al., 2018. Shows variation on model simulated
sulfate deposition for the 1815 Tambora eruption.

L380: Maybe add: … with the bipolar ratio of simulated sulfate depositions between northern and …

Added.

L384-387: Here would be a good opportunity to put your results into context with a recent study
(Guillet et al., 2023) using another aerosol-climate model (IPSL), tree rings and documentary
evidence of lunar eclipses to constrain the age of the Samalas eruption (May to August 1257 is their
most likely age).

This is a good suggestion, we have added the following text on lines 431-439:

"*Overall, the multi-proxy to model comparison utilized in this study provides a clear distinction between*
*JULY1257 and JAN1258 eruption scenarios, with better agreement between proxy reconstructions and*
*model-simulated anomalies being shown for a July 1257 eruption date. This is consistent with the May-*
*August 1257 date constraint suggested by Guillet et al., (2023) based on their analysis of contemporary*
*reports of total lunar eclipses, combined with tree ring-based climate proxies and aerosol model*
*simulations. Although consensus has converged on a Summer 1257 date for the Samalas eruption, it*
*remains to be seen if a more precise constraint (i.e. to a specific month) could be achieved given current*
*model and proxy uncertainties (as discussed in Section 4.3 below). Nonetheless, this four-month*
*window remains an improvement upon previous dating uncertainty and is still sufficient at present for*
*interrogating the both climatic and human consequences following the eruption.*"

L407: A stronger simulated cooling in the southern hemisphere (SH) must be expected since your model favors aerosol transport to the SH. With the absence of abundant well dated climate proxies for the SH (apart from ice cores in Antarctica) it will remain impossible to resolve the model/proxy disagreement previously discussed for the SH (Neukom et al., 2014).

The lack of well dated, geographically distributed climate proxies in the SH is a frustrating limitation. This hinders both attempts to resolve model/proxy disagreements and attempts to understand the climatic (and societal) impacts following the eruption. We have revised the sentence in question to acknowledge stronger simulated cooling in the SH is as expected for a model that favours aerosol transport to the SH, although differences in land/sea extend between hemisphere likely also contributes to asymmetric cooling.

*Lines 464-469: "Instead, model-simulated SAT anomalies for a July 1257 eruption only show significant negative SAT anomalies across South America, Africa, and Oceania beginning in late summer 1257, with cooling in some regions of up to -2°C, although without well-dated, geographically distributed climate proxies in the SH it will remain difficult to resolve whether this is potential model/proxy discrepancy in the SH (Neukom et al., 2014). However, if extreme and sudden cooling did occur in the SH, it would be expected to have significant, but as yet unknown, consequences for communities and civilizations in the Southern Hemisphere."*

L410-411: Since you invoke a potential connection between potential post-volcanic climate change and migration in the 13th century you may want to expand this to include two more large volcanic eruptions during this period in 1269 and 1275 both suspected to be located in the SH.

We agree that it is highly plausible that these later eruptions and their associated climate perturbations may have led to additional adverse effects on impacted civilisations, especially given the short windows available between events for recovery. We have therefore added the following paragraph on lines 469-474:

*"Additional large volcanic eruptions in 1269, 1278, and 1286, which combined with the eruption in 1257, make the sulfate loading in the 13th century two to ten times larger than any other century in the last 1500 years (Gao et al., 2008), and may have led to further cooling of climate with the effects on impacted civilisations being prolonged throughout the latter half of the 13th century. For example, the first settlement of New Zealand most likely occurred between 1250–1275, with suggestions this may have reflected a climate-induced migration associated at least in part with the impacts of the Mt. Samalas, and subsequent, eruptions (Anderson, 2016, Bunbury et al., 2022)."*

L432: Please consider removing this idea put forward on chronological errors in tree-ring chronologies or provide the right balance by also citing the numerous studies that have univocally rejected this speculation on globally missing tree rings (Anchukaitis et al., 2012; Buntgen et al., 2014; Esper et al., 2013; St. George et al., 2013).

This idea has been removed.

L438: Stoffel et al. (2015) have reconstructed temperatures for the past 1,500 years and largely focused on the Samalas 1257 and Tambora 1815 eruptions. Other studies have extended the focus to the Common Era (Büntgen et al., 2020; Büntgen et al., 2016; Sigl et al., 2015)

These references have been added.

L459/462: Annual resolution alone is not important; annual dating precision is the key requirement of a proxy in order to analyze volcanic climate impacts.

The inclusion of hydroclimate anomalies has been removed given the limited number of ensemble
realisations and appropriate proxy records. We felt that the scope and robustness of the paper was
more streamlined without inclusion of this work.

L466-470: A strong asymmetric distribution of sulfate is however not supported by the numerous
ice-core records which show quite comparable magnitudes in deposition following Samalas and
other known tropical eruptions (e.g. Tambora, Cosiguina, Krakatau) independent of their eruption
season. (Table 1) How consistent is this interhemispheric asymmetry for different seasons across
aerosol-climate models?

Ice core records show a mean Greenland/Antarctica sulfate deposition ratio of 1.4  for the Samalas
eruption which does suggest a degree of asymmetry in hemispheric sulfate deposition. Similarly other
known tropical eruptions show asymmetric distribution - e.g Tambora (0.8), Cosigina (2), and Krakatau
(1.7) - from Toohey and Sigl, 2017. The asymmetric distribution shown by Samalas is most similar to
Krakatau where the two eruptions also have the most similar latitudes and seasonal timing. Therefore,
variation in the direction and magnitude of asymmetric sulfate distribution most likely reflects the
different eruption latitudes, seasons, and atmospheric conditions (e.g QBO phase).

Marshall et al., 2018 showed considerable inter-model variation in the hemispheric distribution and
sulfate deposition for the Tambora 1815 eruption (see Table S3) where the Greenland/Antarctica ratio
varies between 0.7 and 3.3. For the Tambora 1815 eruption, UM-UKCA (an earlier version of the
UKESM) results in greater deposition in Greenland whilst for our simulations of the Samalas 1257
eruption using the UKESM for both July and January ensembles deposition is greater in Antarctica.
Therefore variation in hemispheric asymmetry for simulated sulfate deposition relfects both inter and
intra-model differences. A paragraph reflecting this has been added to the updated discussion section
on lines 549 - 568 and as outlined in our response to the comment on L347-358 on page 8 above.

L 475: High temporal resolution is readily available for ice core records and volcanic eruptions
located close-by or upwind of the ice sheets are already dated to the season using these ice cores
(Veidivötn 1476/77, Paektu 946/47, Eldgja spring 939, Churchill 852/53, Okmok II 44/43 BCE; (Abbott
et al., 2021; Mackay et al., 2022; McConnell et al., 2020; Oppenheimer et al., 2018)). The main
limitation to estimate the timing of tropical eruptions stems from the limited number of
observations from known tropical eruptions (see Table 1), variability in the transport time (1-8
months), and the disagreement in aerosol-climate models regarding stratospheric aerosol transport
times between the tropics and the polar regions (see Marshall et al., 2018; Clyne et al. 2021).

The discussion section has been rewritten and line 475 has been removed.

L491-496: You may want to discuss here not only the potential but also the risks of using the
agreement of model simulation output and proxies to constrain source parameters used as original
forcing input; as one might easily run into a conformation bias, where one tends to favor solutions
that best agree with our expectations.

The widely rejected "missing tree-ring hypothesis" (Mann et al., 2012) had its origin in a seemingly
mismatch between climate simulations and proxy reconstructions which had subsequently been
resolved (see PAGES 2k Consortium 2019, Nature Geoscience) by improved estimates of volcanic
forcing (Toohey & Sigl 2007) and better climate reconstructions with inclusions of more tree-ring
MXD records (Stoffel et al., 2015; Schneider et al. 2015; Wilson et al., 2016; Anchukaitis et al. 2017).
In a similar way, the accuracy of ice-core dating has also been challenged because of an apparent
mismatch between strong tree-ring indicated cooling in the high-latitudes of the NH in 1453, and
strong global volcanic forcing in 1458 (Esper et al., 2017). This hypothesis has also been rejected by recognizing evidence for two major eruptions 5 years apart and constraining the date of the larger
signal in Greenland using tephra from an historic eruption in Iceland occurring in 1477 (Abbott et al.,
2021). Both examples highlight the risk of using model-data agreement as the only diagnostic tracer
for the correctness of the underlying records.

We acknowledge the risks of the model-proxy approach, and thank the reviewer for highlighting
these specific examples. An additional paragraph has been added to the discussion to reflect this on
lines 569 to 584 in the revised manuscript. See also our response to the comment on rationale
above.

L498-503: I would omit or rephrase this section. The VSSI as reconstructed by ice cores is free of any
climate response constraints, unlike for example previous reconstructions that had tuned the dates
of volcanic forcing reconstructions (i.e. in 1453) to tree-ring indicated cooling extremes (Gao et al.,
2008; Gao et al., 2006). Moreover, since the VSSI is in Toohey & Sigl (2017) based on 17 individual ice
cores from both hemispheres a bias in the magnitude is much less an issue as for reconstructions
that are based on single ice cores (Bader et al., 2020; Kobashi et al., 2017; Zielinski et al., 1994).
Sampling stratospheric sulfate deposition in ice cores from both hemispheres simultaneously, is
ideal for obtaining representative estimates of magnitude despite possible asymmetric sulfate
distribution. To my knowledge, there are no known well dated large volcanic eruptions from the
tropics which would not have been recognized as such in the polar ice-core records.

If in the future, existing disagreements in aerosol-climate models regarding the global spread of
sulfate aerosols can be resolved it may eventually become possible to further constrain together
with the existing high-resolution ice core records important source parameters such as the eruption
season, latitude and/or plume heights. In any case it appears necessary to cite key studies analyzing
the fate of stratospheric sulfur following major eruptions across the models (Clyne et al., 2021;
Marshall et al., 2018; Quaglia et al., 2023).

The discussion has been comprehensively rewritten and the section on VSSI uncertainty has been
omitted. Several paragraphs have been added focussing on inter-model differences in global sulfate
distribution between global aerosol models:

*Lines 512-521:* *"Model-simulated anomalies are strongly dependent on model set up, including model*
*resolution, modelled stratospheric winds, aerosol microphysics and sedimentation and deposition*
*schemes (Marshall et al., 2018, Quaglia et al., 2023). In recent model intercomparison studies*
*(Marshall et al., 2018, Quaglia et al., 2023) UM-UKCA, a previous version of the UKESM, showed a bias*
*towards stronger transport to the NH extratropics, resulting in a hemispherically asymmetric aerosol*
*load. The spatial distribution of volcanic forcing can influence subsequent growth of sulfate aerosols*
*and their global distribution, in turn affecting the persistence of aerosols in the stratosphere (Quaglia*
*et al., 2023). Compared to other global aerosol models UM-UKCA also has relatively weaker poleward*
*transport, with stronger meridional deposition which may lead to a more equatorially focussed aerosol*
*distribution and deposition (Marshall et al., 2018). Disentangling large inter-model differences from*
*the range of model components that contribute to this uncertainty remains challenging, although*
*future multi-model multi-proxy studies may be of use."*

Lines 549-568: "Nonetheless, model simulations of the Tambora 1815 eruption by Marshall et al., 2018
found a strong model dependency for the timing, magnitude, and spatial distribution of sulfate
deposition. Polar sulfate deposition in a previous version of the UK climate model (UM-UKCA) was half
that reconstructed using ice cores for the Tambora 1815 eruption and considerably lower than
deposition in the other MAECHAM5-HAM and SOCOL-AER models analysed (See Table S3). In our

JUL1257 and JAN1258 simulations for the Samalas 1257 eruption mean total deposition in Antarctica
(55-28 Kg km-2) and Greenland (26-23 Kg km-2) was also lower than mean total deposition from ice
cores records (73 and 105 Kg km-2  respectively, see Table 2). The overall lower polar total sulfate
deposition in the UKESM may be caused by too weak polarward transport (or stronger meridonal
deposition) (Marshall et al., 2018). For our JUL1257 and JAN1258 model-simulated sulfate deposition
for both eruption scenarios result in Greenland to Antarctica ratios < 1, with a Summer 1257 eruption
showing a much more asymmetric distribution (0.49) than a January 1258 eruption (0.83). These ratios
however, disagree with the Greenland to Antarctica ratio derived from ice cores (1.4, Toohey and Sigl,
2017) which if the ice core values are robust, suggests that for our Samalas scenarios, sulfate aerosol
transport to the NH is too low, thus favouring transport to the SH. Of the four models analysed in
Marshall et al., (2018) for the Tambora 1815 eruption, UM-UKCA was one of two models that had
greater deposition in Greenland compared to Antarctica (Table S3, Greenland/Antarctica = 1.7). Both
the Tambora 1815 and Mt Samalas 1257 were large magnitude eruptions at a similar latitude,
therefore this intra-model difference in the asymmetric distribution of sulfate aerosol most likely
results from differences in initial conditions used for our simulations (such as the phase of the QBO).
Overall, this further highlights the complications of disentangling inter-model differences and intra-
model variation due to initial conditions. Given both these inter and intra-model differences in relative
hemispheric aerosol distribution and deposition, the current robustness of using hemispheric sulfate
deposition ratios to distinguish between eruption scenarios when compared to ice core records is
limited, with further work being needed to understand model and starting condition specific effects.
"

**Additional Literature:**

Abbott, P. M., Plunkett, G., Corona, C., Chellman, N. J., McConnell, J. R., Pilcher, J. R., Stoffel, M., and
Sigl, M.: Cryptotephra from the Icelandic Veiðivötn 1477 CE eruption in a Greenland ice core:
confirming the dating of volcanic events in the 1450s and assessing the eruption's climatic impact,
Clim. Past, 17, 565-585, 2021.

Anchukaitis, K. J., Breitenmoser, P., Briffa, K. R., Buchwal, A., Buntgen, U., Cook, E. R., D'Arrigo, R. D.,
Esper, J., Evans, M. N., Frank, D., Grudd, H., Gunnarson, B. E., Hughes, M. K., Kirdyanov, A. V., Korner,
C., Krusic, P. J., Luckman, B., Melvin, T. M., Salzer, M. W., Shashkin, A. V., Timmreck, C., Vaganov, E.
A., and Wilson, R. J. S.: Tree rings and volcanic cooling, Nat Geosci, 5, 836-837, 2012.

Bader, J., Jungclaus, J., Krivova, N., Lorenz, S., Maycock, A., Raddatz, T., Schmidt, H., Toohey, M., Wu,
C.-J., and Claussen, M.: Global temperature modes shed light on the Holocene temperature
conundrum, Nat Commun, 11, 4726, 2020.

Büntgen, U., Arseneault, D., Boucher, É., Churakova, O. V., Gennaretti, F., Crivellaro, A., Hughes, M.
K., Kirdyanov, A. V., Klippel, L., Krusic, P. J., Linderholm, H. W., Ljungqvist, F. C., Ludescher, J.,
McCormick, M., Myglan, V. S., Nicolussi, K., Piermattei, A., Oppenheimer, C., Reinig, F., Sigl, M.,
Vaganov, E. A., and Esper, J.: Prominent role of volcanism in Common Era climate variability and
human history, Dendrochronologia, 64, 125757, 2020.

Büntgen, U., Myglan, V. S., Ljungqvist, F. C., McCormick, M., Di Cosmo, N., Sigl, M., Jungclaus, J.,
Wagner, S., Krusic, P. J., Esper, J., Kaplan, J. O., de Vaan, M. A. C., Luterbacher, J., Wacker, L., Tegel,
W., and Kirdyanov, A. V.: Cooling and societal change during the Late Antique Little Ice Age from 536
to around 660 AD, Nat Geosci, 9, 231-236, 2016.

Büntgen, U., Wacker, L., Nicolussi, K., Sigl, M., Gutler, D., Tegel, W., Krusic, P. J., and Esper, J.:
Extraterrestrial confirmation of tree-ring dating, Nat Clim Change, 4, 404-405, 2014.

Clyne, M., Lamarque, J. F., Mills, M. J., Khodri, M., Ball, W., Bekki, S., Dhomse, S. S., Lebas, N., Mann,
G., Marshall, L., Niemeier, U., Poulain, V., Robock, A., Rozanov, E., Schmidt, A., Stenke, A.,
Sukhodolov, T., Timmreck, C., Toohey, M., Tummon, F., Zanchettin, D., Zhu, Y., and Toon, O. B.:
Model physics and chemistry causing intermodel disagreement within the VolMIP-Tambora
Interactive Stratospheric Aerosol ensemble, Atmos. Chem. Phys., 21, 3317-3343, 2021.

Esper, J., Buntgen, U., Hartl-Meier, C., Oppenheimer, C., and Schneider, L.: Northern Hemisphere
temperature anomalies during the 1450s period of ambiguous volcanic forcing, B Volcanol, 79, 2017.

Esper, J., Buntgen, U., Luterbacher, J., and Krusic, P. J.: Testing the hypothesis of post-volcanic
missing rings in temperature sensitive dendrochronological data, Dendrochronologia, 31, 216-222,
2013.

Gao, C. C., Robock, A., and Ammann, C.: Volcanic forcing of climate over the past 1500 years: An
improved ice core-based index for climate models, J Geophys Res-Atmos, 113, 2008.

Gao, C. C., Robock, A., Self, S., Witter, J. B., Steffenson, J. P., Clausen, H. B., Siggaard-Andersen, M. L.,
Johnsen, S., Mayewski, P. A., and Ammann, C.: The 1452 or 1453 AD Kuwae eruption signal derived
from multiple ice core records: Greatest volcanic sulfate event of the past 700 years, J Geophys Res-
Atmos, 111, 2006.

Guillet, S., Corona, C., Oppenheimer, C., Lavigne, F., Khodri, M., Ludlow, F., Sigl, M., Toohey, M.,
Atkins, P. S., Yang, Z., Muranaka, T., Horikawa, N., and Stoffel, M.: Lunar eclipses illuminate timing
and climate impact of medieval volcanism, Nature, 616, 90-95, 2023.

Jong, L. M., Plummer, C. T., Roberts, J. L., Moy, A. D., Curran, M. A. J., Vance, T. R., Pedro, J. B., Long,
C. A., Nation, M., Mayewski, P. A., and van Ommen, T. D.: 2000 years of annual ice core data from
Law Dome, East Antarctica, Earth Syst. Sci. Data, 14, 3313-3328, 2022.

Kobashi, T., Menviel, L., Jeltsch-Thommes, A., Vinther, B. M., Box, J. E., Muscheler, R., Nakaegawa, T.,
Pfister, P. L., Doring, M., Leuenberger, M., Wanner, H., and Ohmura, A.: Volcanic influence on
centennial to millennial Holocene Greenland temperature change, Sci Rep-Uk, 7, 2017.

Mackay, H., Plunkett, G., Jensen, B. J. L., Aubry, T. J., Corona, C., Kim, W. M., Toohey, M., Sigl, M.,
Stoffel, M., Anchukaitis, K. J., Raible, C., Bolton, M. S. M., Manning, J. G., Newfield, T. P., Di Cosmo,
N., Ludlow, F., Kostick, C., Yang, Z., Coyle McClung, L., Amesbury, M., Monteath, A., Hughes, P. D. M.,
Langdon, P. G., Charman, D., Booth, R., Davies, K. L., Blundell, A., and Swindles, G. T.: The
852/3 CE Mount Churchill eruption: examining the potential climatic and societal impacts
and the timing of the Medieval Climate Anomaly in the North Atlantic region, Clim. Past, 18, 1475-
1508, 2022.

Marshall, L., Schmidt, A., Toohey, M., Carslaw, K. S., Mann, G. W., Sigl, M., Khodri, M., Timmreck, C.,
Zanchettin, D., Ball, W. T., Bekki, S., Brooke, J. S. A., Dhomse, S., Johnson, C., Lamarque, J. F.,
LeGrande, A. N., Mills, M. J., Niemeier, U., Pope, J. O., Poulain, V., Robock, A., Rozanov, E., Stenke, A.,
Sukhodolov,T., Tilmes, S., Tsigaridis, K., and Tummon, F.: Multi-model comparison of the volcanic
sulfate deposition from the 1815 eruption of Mt. Tambora, Atmos. Chem. Phys., 18, 2307-2328,
2018.

McConnell, J. R., Edwards, R., Kok, G. L., Flanner, M. G., Zender, C. S., Saltzman, E. S., Banta, J. R.,
Pasteris, D. R., Carter, M. M., and Kahl, J. D. W.: 20th-century industrial black carbon emissions
altered arctic climate forcing, Science, 317, 1381-1384, 2007.

McConnell, J. R., Sigl, M., Plunkett, G., Burke, A., Kim, W. M., Raible, C. C., Wilson, A. I., Manning, J. G., Ludlow, F., Chellman, N. J., Innes, H. M., Yang, Z., Larsen, J. F., Schaefer, J. R., Kipfstuhl, S., Mojtabavi, S., Wilhelms, F., Opel, T., Meyer, H., and Steffensen, J. P.: Extreme climate after massive eruption of Alaska's Okmok volcano in 43 BCE and effects on the late Roman Republic and Ptolemaic Kingdom, P Natl Acad Sci USA, 117, 15443-15449, 2020.

Neukom, R., Gergis, J., Karoly, D. J., Wanner, H., Curran, M., Elbert, J., Gonzalez-Rouco, F., Linsley, B. K., Moy, A. D., Mundo, I., Raible, C. C., Steig, E. J., van Ommen, T., Vance, T., Villalba, R., Zinke, J., and Frank, D.: Inter-hemispheric temperature variability over the past millennium, Nat Clim Change, 4, 362-367, 2014.

Oppenheimer, C., Orchard, A., Stoffel, M., Newfield, T. P., Guillet, S., Corona, C., Sigl, M., Di Cosmo, N., and Büntgen, U.: The Eldgja eruption: timing, long-range impacts and influence on the Christianisation of Iceland, Climatic Change, 147, 369-381, 2018.

PAGES2k Consortium: Data Descriptor: A global multiproxy database for temperature reconstructions of the Common Era, Sci Data, 4, 2017.

Plummer, C. T., Curran, M. A. J., van Ommen, T. D., Rasmussen, S. O., Moy, A. D., Vance, T. R., Clausen, H. B., Vinther, B. M., and Mayewski, P. A.: An independently dated 2000-yr volcanic record from Law Dome, East Antarctica, including a new perspective on the dating of the 1450s CE eruption of Kuwae, Vanuatu, Clim Past, 8, 1929-1940, 2012.

Quaglia, I., Timmreck, C., Niemeier, U., Visioni, D., Pitari, G., Brodowsky, C., Brühl, C., Dhomse, S. S., Franke, H., Laakso, A., Mann, G. W., Rozanov, E., and Sukhodolov, T.: Interactive stratospheric aerosol models' response to different amounts and altitudes of SO2 injection during the 1991 Pinatubo eruption, Atmos. Chem. Phys., 23, 921-948, 2023.

Salzer, M. W. and Hughes, M. K.: Bristlecone pine tree rings and volcanic eruptions over the last 5000 yr, Quaternary Res, 67, 57-68, 2007.

Sigl, M., McConnell, J. R., Layman, L., Maselli, O., McGwire, K., Pasteris, D., Dahl-Jensen, D., Steffensen, J. P., Vinther, B., Edwards, R., Mulvaney, R., and Kipfstuhl, S.: A new bipolar ice core record of volcanism from WAIS Divide and NEEM and implications for climate forcing of the last 2000 years, J Geophys Res-Atmos, 118, 1151-1169, 2013.

Sigl, M., McConnell, J. R., Toohey, M., Curran, M., Das, S. B., Edwards, R., Isaksson, E., Kawamura, K., Kipfstuhl, S., Kruger, K., Layman, L., Maselli, O. J., Motizuki, Y., Motoyama, H., Pasteris, D. R., and Severi, M.: Insights from Antarctica on volcanic forcing during the Common Era, Nat Clim Change, 4, 693-697, 2014.

Sigl, M., Winstrup, M., McConnell, J. R., Welten, K. C., Plunkett, G., Ludlow, F., Büntgen, U., Caffee, M., Chellman, N., Dahl-Jensen, D., Fischer, H., Kipfstuhl, S., Kostick, C., Maselli, O. J., Mekhaldi, F., Mulvaney,R., Muscheler, R., Pasteris, D. R., Pilcher, J. R., Salzer, M., Schupbach, S., Steffensen, J. P., Vinther, B. M., and Woodruff, T. E.: Timing and climate forcing of volcanic eruptions for the past 2,500 years, Nature, 523, 543-549, 2015.

St. George, S., Ault, T. R., and Torbenson, M. C. A.: The rarity of absent growth rings in Northern Hemisphere forests outside the American Southwest, Geophys Res Lett, 40, 3727-3731, 2013.

Zielinski, G. A., Mayewski, P. A., Meeker, L. D., Whitlow, S., Twickler, M. S., Morrison, M., Meese, D. A., Gow, A. J., and Alley, R. B.: Record of Volcanism since 7000-Bc from the Gisp2 Greenland Ice Core and Implications for the Volcano-Climate System, Science, 264, 948-952, 1994.

**Reviewer 2**

The authors present a multi-proxy to model comparison study of the Mt. Samalas eruption, the
largest explosive sulfur-rich eruptions of the last millennium, which eruption season/year is still not
known. As potential eruption dates NH summer 1257 and early 1258 are discussed. To achieve a
more-precise constraint of the year and season of the Mt. Samalas eruption, the authors run
ensemble simulations with the UK Earth System Model (UKSEM1) for a range of eruption scenarios
and initial conditions for a NH summer and winter eruption and compare them with spatially
resolved multi-proxy data. This allows them to robustly distinguish between both eruption dates.
The authors suggest July 1257 as the most likely initial date due to its better agreement with
spatially averaged and regionally resolved proxy surface temperature reconstructions. Overall, it is
solid piece of work and important for further applications, but needs some clarifications and
improvements. I therefore recommend publication after revisions, see below.

We thank reviewer 2 for their comprehensive comments and helpful suggestions on our manuscript.
Please find our response to your comments below:

**General comments**

In my opinion, the discussion part needs some revision as some important points are not mention at
all or only briefly touched.

I miss a dedicated paragraph about volcanic forcing uncertainties. The authors mention in one
sentence that there might be uncertainties in the VSSI estimate. There was recently a study
published in Climate of the Past by Lücke et al. (2023) who addressed the effect of uncertainties in
natural forcing records on simulated temperature during the last millennium with a volcanic forcing
ensemble. In Lücke et al. (2023) also the large uncertainties around the Samals eruption were
addressed, thus it would be good to discuss your results with respect to their work.

Timmreck et al. (2021) also discussed forcing uncertainties in comparison with multiple-proxy data
for the 1809 eruption showing that NH large-scale climate modes are sensitive to both volcanic
forcing strength and its spatial structure.  As the spatial structure of the forcing pattern is quite
important, I wonder, if the spatial volcanic forcing distribution is similar for the different realizations
of each starting date and how does it differ between them. Observations show that some tropical
eruptions had a hemispherical asymmetric aerosol load e.g. Agung 1963 or El Chichon 1982. The
spatial structure might also be a potential source of uncertainty and should be addressed in the
discussion section.

A more comprehensive discussion of volcanic forcing uncertainties has been added to the discussion
under 4.3 Model and Proxy Limitations.

*Lines 524-542: "Even within a single model, uncertainties persist in initial eruption conditions (e.g*
*phase of the QBO,) and volcanic forcing parameters (e.g timing, magnitude, injection height and*
*latitude of eruption). Timmerick et al., (2021) demonstrated that both the magnitude of forcing as well*
*as its spatial structure can similarly affect proxy–simulation comparisons, particularly in the NH*
*extratropics. This is supported by Lücke et al., (2023) who demonstrate a significant spread in the*
*temperature response due to volcanic forcing uncertainties which can strongly affect the agreement*
*with proxy reconstructions. The VSSI estimate used in our study was taken from the eVolv2k*
*reconstruction (Toohey and Sigl, 2017) and is within error of other $SO_2$ emission estimates for the Mt*
*Samalas eruption (Vidal et al., 2016, Lavigne et al., 2013). Whilst maximum plume heights have been*
*estimated for the Samalas eruption (~ 43 km Vidal et al., 2015) the $SO_2$ injection height remains*

*unknown. In our simulations the injection height is set to 18-20 km, which may be too low, but does allow for lofting of sulfate aerosol. Moreover, Stoffel et al., (2015) found that increasing plume height from 22-26 km to 33-36 km for simulations of the Mt Samalas 1257 eruption increased the magnitude of the peak post-eruption NH JJA temperature anomaly to -4°C for a January eruption and -1°C and -2°C for May and July eruption scenarios respectively. A plume height greater than the 20km used in our study would therefore likely further enforce our central conclusion that better agreement between proxy and model-simulated temperature anomalies is achieved for a summer 1257 eruption date, whilst a greater plume height for a January 1258 eruption would only further overpredict the magnitude of post-eruption cooling. Timmreck et al., (2009) also highlight the strong dependence of model-simulated post-eruption climate responses following large volcanic eruptions on the aerosol particle size distribution due to self-limiting effects of larger particles (Pinto et al., 1989). These particle characteristics are difficult to constrain retrospectively for historical eruptions such as Mt. Samalas and thus represent a significant uncertainty when simulating historical eruptions."*

I also miss in the discussion a dedicated paragraph about the strength and the weaknesses of the applied global aerosol model. The recent global aerosol model intercomparison studies (Marshall et al. 2018, Clyne et al. 2021, Quaglia et al. 2023) reveal several difficulties, which the current generation of global aerosol model has to face too. Marshall et al. (2018) demonstrate for example that the ratio of the hemispheric atmospheric sulfate aerosol burden after the eruption to the average ice sheet deposited sulfate varies between models by up to a factor of 15. The study by Qualia et al. (2023) where the different model results are compared to satellite observations after the Pinatubo episode show a stronger transport towards the NH extratropics, suggesting a much weaker subtropical barrier in all the models. Hence, I wonder how model specific are your results? How much are the results presented here influenced by biases or specific features of the UKESM model. Would not a multi-model multi-proxy intercomparison be the best suitable way to move forward?

A fuller discussion of the specific strengths and weaknesses of the UKESM with respect to other global aerosol models have been added to the discussion under 4.3 Model and Proxy Limitations, particularly with respect to hemispheric aerosol distribution (see Table S3, reproduced from Marshall et al., 2018). Further multi-model intercomparision studies would undoubtedly be useful, especially when compared to proxy constraints. Given the nature of this project (1-year MSci) incorporating additional models was beyond its scope. However, we hope further studies may compare alternative models to our results using the UKESM.

We have added an additional paragraph to the final manuscript (lines 549-568). Please also see our response to reviewer 1 (lines 275-313 in this document) above.

I wonder why you run only a July and a January scenario and not an experiment for the autumn season. Toohey at al. (2011) demonstrate that the modulation by the annual cycle for many variables is not linear. An experiment with the initial date at the 1st of October could have been a very valuable set up.

Whilst additional model simulations for a Spring or Autumn eruption date would certainly be valuable, it is not possible to run any further simulations as part of this project. Summer and winter realisations were chosen as end-member scenarios to emphasise the effects of annual variation in atmospheric circulation. The suggested dates for the Samalas eruption specifically also fall into either Summer or Winter eruption scenarios, and so given model resources these were the priority to test.

**Specific comments**

Lines 18 ff.: The description of the initialization of the volcanic cloud misses some important details.
For me it is not clear, how you initialize your volcanic cloud on the horizontal grid. Do you inject your
sulfur emission in one grid box around the location of the volcano or over several grid boxes or even
in a zonal band at 8 S. As shown by Quaglia et al. (2023), the results could be very different for the
UKESM depend on the initialization of the eruption cloud. Please, give some more details here and
also modify Table 1 accordingly as "8 S" is a bit unspecific in the respect.

The $SO_2$ is injected into the grid boxes at 8 S in the altitude range (18-20 km), so no horizontal
spreading, but vertically. Table 1 has been modified to make this clearer.

Lines 45-46 Please add references.

This section has been rewritten:

*Lines 46-49:* "The full span of dates proposed for the Mt Samalas eruption ranges from 1256 to 1258,
*with suggestions including an eruption in spring 1256 (Bauch, 2019), summer 1257 (Lavigne et al.,*
*2013, Oppenheimer 2003), and early 1258 (Stothers 2000). Whilst consensus has converged on a*
*summer 1257 eruption date (Guillet et al., 2023, Büntgen et al., 2022), as of yet, no single combination*
*of evidence has focussed on distinguishing between, and excluding other dates proposed for the Mt*
*Samalas eruption."*

Line 200 : As you discuss also in 3.1.2 only the NH data, it might be appropriate to change the
subsection title to "NH hemispheric mean" or something along this line.

The subsections in Section 3 are currently titled:

3.1.1 Northern Hemisphere Model and Tree Ring Constraints

3.1.2 Globally-Resolved Model and Multi-Proxy Constraints.

Section 3.1.2 now includes 4 additional ice core records from Antarctica and the tree ring records in
Australia and New Zealand will be discussed, therefore it will be "Globally resolved".

Line 200 ff.: I wonder a bit why you calculate your own uncertainties for the tree ring reconstruction
and do not use the ensemble spread of tree ring ensemble reconstruction from Büntgen et al.
(2021), see for example Figure 6 in van Dijk et al (2022).

The tree ring mean that is shown in Figure 1 is calculated as the mean of four tree ring ensemble
reconstructions from: Wilson et al., (2016), Schneider et al., (2015), Büntgen et al., (2021), Guillet et
al., (2017). Therefore, we calculate our own uncertainties to account for the combination of the four
reconstructions. See figure S1 for a comparison of the four different records.

Line 201 and elsewhere: I suggest that you give the two experiments dedicated names e.g JUN1257
or JAN1258 to avoid confusion by just saying the date.

Done.

Line 418 : How many individual realizations have a positive ENSO phase in summer 1258 and 1259?
You can also look to relative SSTs instead of raw SSTs here.

Across the individual realisations, 5 have positive SST anomalies in the ENSO 3.4 index region (170-
120°W, -5-5°N) for 1258-59. These realisations are only classified as having positive SST anomalies as
a fuller ENSO classification would require analysis of relative SST anomalies. An investigation of volcano-ENSO interactions is beyond the scope of this work and this sentence was just to
acknowledge there is some dependence on initial ENSO conditions.

Line 450: Does a best estimate for the emission height really exist?

The phrasing "current best estimate" was intended in the sense that with the information we currently
have, this is the best estimate we can produce at this time based on that evidence. We didn't intend
to imply these were absolutely the best estimates and so have removed the word "best":

Lines 528-530: "*The VSSI estimate used in our study was taken from the eVolv2k reconstruction
(Toohey and Sigl, 2017) and is within error of other SO2 emission estimates for the Mt Samalas
eruption (Vidal et al., 2016, Lavigne et al., 2013).*"

Lines 459-60: Reference is missing

This paragraph has been rewritten (see response to comment on Line 462 below).

Lines 462: Not clear to me. According to their analysis of speleothem data from Mesoamerica, Ridley
et al (2015) showed that SH volcanic eruptions, including those at low southerly latitudes (e.g.
Tambora 1815) force the ITCZ to the north and lead to wetter conditions. Your figure S11 shows for
Mexico a similar response for Tambora. Tejedor et al. (2021) showed on the other hand results for a
super epoch analysis.

This paragraph has been removed as we felt the scope of the paper was more streamlined without
the inclusion of hydroclimate anomalies. Whilst their inclusion would certainly be valuable, in order
to do so robustly would require more ensemble realisations and additional appropriate proxy
records.

Line 491ff:: You should not forget to discuss the model deficits in this paragraph; nine realizations
might not be a sufficient number for each model experiment to obtain statistically significant pattern
of tropical hydroclimate changes, large scale meridional transport and sulfate deposition are  also
strongly model dependent, see Marshall et al. (2018), Quaglia et al. (2023)

As above, the section on hydroclimate anomalies has been removed.

Line 495: Another exemplary study in this respect is the paper by van Dijk et al. (2023) which you
could cite here as well

This reference has been added (throughout where relevant).

Lines 780 ff.: References of Wade is listed twice, also indicated as 2020a and 2020b

This has been corrected.

**Figures:**

Figure 2: Maybe you include here in one of the panels the specific position of the tree-rings.

We tested including the positions of the specific trees rings used in the N-TREND reconstruction on
Figure 2, however, we felt that this made the plot look cluttered and obscured the temperature
contours (which are central to the plot). Therefore, an additional figure has been added to the
Supplementary showing the locations of the records used in the N-TREND reconstruction:

[Figure]

**"Figure S3: Spatial distribution of sites used in the N-TREND reconstruction, with proxy type**
**shown by colour.** N-TREND data from Wilson et al., (2016) and Anchukaitis et al., (2017) , plot after
Fig 1 in Wilson et al., (2016)."

Figure 3: Difficult to interpret the proxies in the two lower rows. The colors in the upper row
probably not refer to the colormap at the bottom, so please use different colors instead of red and
blue here. Which meaning has the cyan color here? I also wonder if it would make sense to show
here only the NH as you do not discuss the two proxies from the SH here. They could be shown in
the supplements.

Figure 3 has been reworked into two separate figures (Figure 3 and Figure S10, see below). The new
Figure 3 in the main body of text includes only the model-multi proxy comparison panels. The proxy
symbol size has been increased. Additional ice core records from Antarctica have also been added.

[Figure]

**"Figure 3: Globally-resolved multi-proxy-model comparison visualized for summers (JJA) 1258 and 1259.**

**Symbols denote proxy data type and red/blue shading shows model-simulated surface air temperature**

**anomalies for JUL1257 eruptions (a-b) and JAN1258 eruptions (c-d) ensemble means. Surface air**

**temperature anomalies were calculated relative to a 10-year background climatology constructed from the**

**control ensemble mean. Hashed lines denote anomalies at <95% significance as determined by a grid point**

**ANOVA analysis. Black filled symbols denote agreement within +/- 1°C between model-simulated**

**anomalies and quantitative proxy records. Grey filled symbols denote qualitative agreement with proxy**

**records. Locations and proxy-constrained SAT anomalies are shown in Figure S3."**

Figure S3 now shows only the proxy locations and if anomalies are warm/cool as denoted by blue/red in-fill where the cyan colour highlights refers specifically to frost ring records.

[Figure]

**"Figure S10: Map showing the locations of records included in the multi-proxy database for summer 1258 and 1259. Red/Blue colouring shows positive or negative SAT anomalies respectively, where light blue shading specifically refers to frost rings (i.e non-quantitative tree ring records). Historical records (square symbol) may only make reference to abnormal weather conditions in one year and therefore may only be present on one panel."**

Figure S2, S3: As the number of individual realizations are not import in the context, I suggest to
combine both figures into a new one with two panels one for each starting date and thin lines for
the individual realizations and a thick one for the ensemble mean.

This figure has been remade with all individual realisations on one panel for each starting date:

*"**Figure S2.** Model-simulated Northern Hemisphere Summer (June-July-August) Temperature Anomalies for individual July 1257 (left) and January 1258 (right) ensembles. Black line shows the tree-ring reconstructed mean with the grey band showing 2σ around the mean. Thin lines show individual ensembles, and the thick line shows the model-simulated ensemble mean. Of the 9 July 1257 ensembles 6 lie within 2σ of the tree ring-reconstructed mean for Summer 1258 whereas for the January 1258 ensembles only 2 lie within 2σ of the tree ring-reconstructed mean for Summer 1258."*

Figure S11: Please list the reference of the reconstruction

The reference has been added.

**Data availability:** Please make sure that the data are available before the submission of the revised version.

Data has been uploaded to the CEDA archive and is pending review. Catalogue record can be found at https://catalogue.ceda.ac.uk/uuid/e0221b37aa174dd290c5e105263b59d1

**Literature**

Büntgen, U., Allen, K., Anchukaitis, K. J., et al.: The influence of decision-making in tree ring-based climate reconstructions, Nat. Commun., 12, 1–10, 2021.

Clyne, M., Lamarque, J.-F., Mills, M. J., Khodri, M., Ball, W., Bekki, S., Dhomse, S. S., Lebas, N., Mann, G., Marshall, L., Niemeier, U., Poulain, V., Robock, A., Rozanov, E., Schmidt, A., Stenke, A., Sukhodolov, T., Timmreck, C., Toohey, M., Tummon, F., Zanchettin, D., Zhu, Y., and Toon, O. B.: Model physics and chemistry causing intermodel disagreement within the VolMIP-Tambora Interactive Stratospheric Aerosol ensemble, Atmos. Chem. Phys., 21, 3317–3343, https://doi.org/10.5194/acp-21-3317-2021, 2021.   -

Marshall, L., Schmidt, A., Toohey, M., Carslaw, K. S., Mann, G. W., Sigl, M., Khodri, M., Timmreck, C., Zanchettin, D., Ball, W. T., Bekki, S., Brooke, J. S. A., Dhomse, S., Johnson, C., Lamarque, J.-F., LeGrande, A. N., Mills, M. J., Niemeier, U., Pope, J. O., Poulain, V., Robock, A., Rozanov, E., Stenke, A., Sukhodolov, T., Tilmes, S., Tsigaridis, K., and Tummon, F.: Multi-model comparison of the volcanic sulfate deposition from the 1815 eruption of Mt. Tambora, Atmos. Chem. Phys., 18, 2307–2328, https://doi.org/10.5194/acp-18-2307-2018, 2018.

Lücke, L. J., Schurer, A. P., Toohey, M., Marshall, L. R., and Hegerl, G. C.: The effect of uncertainties in natural forcing records on simulated temperature during the last millennium, Clim. Past, 19, 959–978, https://doi.org/10.5194/cp-19-959-2023, 2023.

Quaglia, I., Timmreck, C., Niemeier, U., Visioni, D., Pitari, G., Brodowsky, C., Brühl, C., Dhomse, S. S., Franke, H., Laakso, A., Mann, G. W., Rozanov, E., and Sukhodolov, T.: Interactive stratospheric aerosol models' response to different amounts and altitudes of $SO_2$ injection during the 1991 Pinatubo eruption, Atmos. Chem. Phys., 23, 921–948, https://doi.org/10.5194/acp-23-921-2023, 2023.

Ridley H, Asmerom Y, Baldini JUL, Breitenbach SFM,Aquino VV,Prufer KM, Culleton BJ, Polyak V, Lechleitner FA, Kennett DJ, Zhang M,Marwan N, Macpherson CG, Baldini LM, Xiao T, PeterkinJL, Awe J, Haug GH. Aerosol forcing of the position of the intertropical convergence zone since AD 1550. Nat Geosci 2015,8: 195–200, doi: 10.1038/ngeo235.

Tejedor, E., Steiger, N.J., Smerdon, J.E., Serrano-Notivoli, R. and Vuille, M. (2021). Global
hydroclimatic response to tropical volcanic eruptions over the last millennium. Proceedings of the
National Academy of Sciences, 118(12), p.e2019145118. doi:10.1073/pnas.2019145118.

Timmreck, C., Toohey, M., Zanchettin, D., Brönnimann, S., Lundstad, E., & Wilson, R. (2021). The
unidentified eruption of 1809: a climatic cold case. Climate of the Past, 17(4), 1455-963 1482.
https://doi.org/10.5194/cp-17-1455-2021.

Toohey, M., Krüger, K., Niemeier, U. and Timmreck, C. (2011). The influence of eruption season on
the global aerosol evolution and radiative impact of tropical volcanic eruptions. Atmospheric
Chemistry and Physics, 11(23), pp.12351–12367. doi:10.5194/acp-11-12351-2011.

van Dijk, E., Jungclaus, J., Lorenz, S., Timmreck, C., and Krüger, K.: Was there a volcanic-induced long-
lasting cooling over the Northern Hemisphere in the mid-6th–7th century?, Clim. Past, 18, 1601–
1623, https://doi.org/10.5194/cp-18-1601-2022, 2022.

van Dijk, E., Mørkestøl Gundersen, I., de Bode, A., Høeg, H., Loftsgarden, K., Iversen, F., Timmreck,
C., Jungclaus, J., and Krüger, K.: Climatic and societal impacts in Scandinavia following the 536 and
540 CE volcanic double event, Clim. Past, 19, 357–398, https://doi.org/10.5194/cp-19-357-2023,
2023.

---

## Referee Report (RR1)

**Utilizing a Multi-Proxy to Model Comparison to Constrain the Season and Regionally Heterogeneous Impacts of the Mt. Samalas 1257 Eruption**

Laura Wainman, Lauren R. Marshall, and Anja Schmidt

The authors have fully addressed the numerous points I had previously raised in their response letter and justified their positions well. The revised version of the manuscript very comprehensively presents the current state of research on this important eruption, including a brief perspective on ongoing research related to the evolution of the plague pathogen in the 13[th] century. It also addresses the opportunities and risks that arise when comparing climate reconstructions from different archives with climate model simulations, owing to uncertainties in the proxies, models and forcing. This is an important contribution to climate research. I spotted a few typos in the manuscript which should be corrected.